# Equity, Justice, and Quality during the COVID-19 Pandemic Period: Considerations on Learning and Scholarly Performance in Brazilian Schools

Ana Dias [1,*], Annibal Scavarda [2], Augusto Reis [1], Haydee Silveira [3] and Ana Scavarda [4]

1   Production Engineering Department, Federal Center for Technological Education Celso Suckow da, Rio de Janeiro 20271-110, Brazil; professor.augusto.reis@gmail.com
2   Production Engineering Department, Federal University of the State of Rio de Janeiro-UNIRIO, Rio de Janeiro 22290-240, Brazil; annibal.scavarda@unirio.br
3   Civil Engineering Department, Federal University of Rio de Janeiro-COPPE-UFRJ, Rio de Janeiro 21941-450, Brazil; haydee.batista@coc.ufrj.br
4   School of Leadership and Education Sciences, University of San Diego, 5998 Alcala Park, San Diego, CA 92110, USA; anacscavarda@gmail.com
*   Correspondence: missdias@gmail.com

**Abstract:** Due to the imperative need for change in habits caused by the COVID-19 pandemic that has plagued the world, this exploratory study plans to analyze the directions taken in teaching activities in public and private schools of the city of Rio de Janeiro (Brazil) and their consequences for learning and scholarly performance concerning elementary and middle schools. In this way, this study verifies through an email questionnaire if there was equality, justice, and quality in teaching methods during the COVID-19 pandemic. The descriptive analysis was carried out based on statistical calculations of quantitative and qualitative variables with various tests, whenever necessary, such as the chi-square, and when inconclusive, Fischer's exact test, Kolmogorov–Smirnov and Shapiro–Wilk, non-parametric Mann–Whitney (when the comparison between two independent groups was mandatory), ANOVA, Kruskal–Wallis, and Friedman test. The results show that teachers tried to interact with students to overcome the problems faced during the COVID-19 pandemic period. Additionally, the study showed that there were differences in scholarly and learning performance, equality, and quality in the types of schools analyzed. This paper will help to fill the literature gap on the subject and will boost ongoing discussion on the inclusion of sustainable concepts in education.

**Keywords:** COVID-19; education; equity; justice; quality

## 1. Introduction

The COVID-19 (SARS-CoV-2 coronavirus) pandemic imposed dramatic changes on daily human activities [1,2] and one of the biggest changes happened in education [3,4]. School closures [5–9] after COVID-19 appearance compelled schools to transition to online programs [10–14]. For Harries et al. [15], the COVID-19 pandemic has affected all levels of education from preschool to post-graduation. All efforts were made to mitigate the spread of the COVID-19 pandemic [16].

Educators all over the world had to find ways to support the needs of their students in an online system [17] and they had to be technically prepared for this endeavor. For Li et al. [18], the teaching method directly impacts the students' learning performance. Moreover, providing online resources became another way of overcoming distance learning barriers [19–21]. Children's learning habits during the COVID-19 period need to be deeply researched [3,4,22–24].

The authors affirm that students' learning habits differ according to social class, aspiration, and educational degree. The combination of affection, humility, optimism, and empathy mitigates online teaching mistakes and risks [25,26]. This exploratory study's

main goal is to observe students' scholarly and learning performance during the COVID-19 pandemic in public and private schools in the city of Rio de Janeiro (Brazil) with respect to elementary and middle schools. It verifies through an email questionnaire if there was equality, justice, and quality in teaching methods during the COVID-19 pandemic. This paper is divided into five sections, including this introductory one. Section 2 describes the theoretical background, Section 3 presents the methods, Section 4 provides the results and discussion, and Section 5 presents the conclusion.

## 2. Theoretical Background

The theoretical section presents the theoretical research background. This paper examines the information put forth in 61 papers gathered through scientific database searches that relate to the themes COVID-19 and education.

*COVID-19 and Education*

During the COVID-19 pandemic period, teachers and authorities tried to provide the best experiences to students. The idea was to overcome social isolation [21,27] and all the unexpected situations and to give adequate support over the COVID-19 pandemic period [28–32]. Social isolation affected not only the students' and professors' health but also the institutions' health [33,34]. Despite the difficulties faced during the COVID-19 pandemic period, schools and universities continued their work [32].

In this sense, the use of technologies in education was the solution found by the authorities to compensate for the lack of face-to-face education [35]. The strategy was successful. The use of technology in education is close to becoming a new normal [3,4,36–38]. For Fischer et al. [38], learning and education should be rethought and reinvented through the integration with innovative technologies, while Lacka et al. [39] believe that technologies are well-integrated with education [40–42]. With online education, the students can obtain a greater amount of information [40,43,44] and are able to build their knowledge base themselves [40].

One of the challenges encountered by the educational system during the COVID-19 pandemic period was the fact that most students do not have the necessary resources to monitor online activities [4,45,46]. Sallaberry et al. [47] highlighted that from the perspective of the professor during the COVID-19 pandemic period, the challenges are related to the reduction of time available to develop the activities, and, in the perception of the student, lack of motivation was outlined as one of the main challenges. The lack of an adequate infrastructure for some students, lack of teacher–student communication and interaction, lack of socialization, impossibility of performing practical applications, and lack of learning motivation are the main problems faced during the COVID-19 pandemic period [48].

Mok el al. [49], Darmody et al. [50] and Andrew et al. [51] affirmed that the COVID-19 pandemic crisis can intensify inequalities in the educational system. Andrew et al. [51] were concerned about inequalities, considering that low-income students have less possibility of accessing the internet and resources available at home such as computers than rich students.

It is evident that the COVID-19 pandemic has affected the right to quality education and the process to guarantee social justice [52]. Studies show a drop in quality when using online methods [53], while Gozzelino et al. [54] and Utunen et al. [55] affirm that it is a considerable challenge to offer quality education during the COVID-19 period. To improve quality education, the following are recommended: improvement of the internet connection and quality of the platforms; education of students' parents on the use of internet devices; and free provision of equipment for low-income students [56]. Moreover, the COVID-19 pandemic period was an opportunity to accelerate the training of high-quality teachers [57].

Education has the objective of promoting social justice and diversity [58]. Social justice is related to the possibility of people's participation, especially the most vulnerable in the education process. Schools and universities must work for social justice, quality, and equity, thus becoming an important tool for social change [48]. Equity means that no one is left behind [59,60].

For Dewart et al. [61], the COVID-19 pandemic will permanently change the educational landscape and it will provide a great lesson in equity, leadership, social justice, and ethics. Torda [62] added that many changes that came with the COVID-19 pandemic will create better educators, collaborators, and innovators, while Tracey and Tolan [63] affirmed that it is an opportunity to explore more effective teaching practices.

### 3. Materials and Methods

In the Methods Section, this paper presents the path taken to achieve its objectives. This research is presented in two parts: the empirical part is based on an email questionnaire sent to schools in the city of Rio de Janeiro (Appendix A), and the theoretical part is based on database searching.

*Procedure and Sample*

This study had as the population the total number of K-12 teachers in the city of Rio de Janeiro. It was based on the 2018 Schools Census (escolas.inf.br, 2018) [64] that registered 66,999 teachers in the city distributed in 2010 private schools, 1439 municipal schools, 457 state schools, and 28 federal schools. To obtain valid results and due to the impossibility of interviewing the entire population, the authors of this paper developed a random sample of the universe. Thus, during October 2020, an email was sent to all schools requesting that teachers answer a questionnaire (Appendix A). The email addresses were obtained through the internet. The results of this study were based on the answers of 116 teachers, 81 (69.2%) females and 35 (30.2%) males, and they were considered significant according to the universe. The research questions were designed to verify if the students' scholarly and learning performance was affected during the COVID-19 pandemic in public and private schools of the city of Rio de Janeiro (Brazil) with respect to elementary and middle schools.

The participants in this research had the following profile: age from 48 to 63 years (56.0%), time of experience well-distributed in the range of 13 to 37 years (94.9%), is a classroom teacher (74.1%), teaches in municipal schools (42.0%), or state schools (35.7%), in high school (38.8%), or in elementary school I (29.3%), and is a Portuguese, English, or Spanish teacher (43.1%). From the collected data, the authors of this paper built a database in an electronic spreadsheet, and it was analyzed by the Statistical Package for the Social Sciences (SPSS) program, version 22.0, and by the Microsoft Excel 2007 application.

For the characterization of the respondent teachers, a variable results descriptive analysis was carried out through frequency distributions with the proportions of interest and calculation of appropriate statistics for quantitative variables (minimum, maximum, average, median, standard deviation, percentiles, and coefficient of variation—CV). The variability in the distribution of a quantitative variable was considered low if CV < 0.20; moderate if $0.20 \leq CV < 0.40$; and high if $CV \geq 0.40$. To check the association between two qualitative variables, comparing the frequency distribution of a qualitative variable in independent groups, the chi-square test was used. When the chi-square test was inconclusive, Fisher's exact test was used instead.

In the inferential analysis of quantitative variables, the hypothesis of normality of the distribution was verified by the Kolmogorov–Smirnov and Shapiro–Wilk tests. The distribution of a variable was considered normal when the two normality tests concluded this way. The student's *t*-test was used for the distributions of two independent groups when the variable followed normal distribution in all groups. For variables that had the hypothesis of normality rejected in at least one of the groups and for ordinal variables, the comparison of two independent groups was performed using the Mann–Whitney nonparametric test. The distributions of a quantitative variable from more than two independent groups were compared by ANOVA when the variable under test followed a normal distribution in all groups, or by the Kruskal–Wallis test when the variable under test did not follow a normal distribution in all groups.

Two repeated measurements from the same respondent were compared in pairs using the Wilcoxon signed posts test, since in all cases at least one of the measurements did not follow a normal distribution. More than two repeated measurements by the same respondent were compared in pairs using the Friedman test, since in all cases at least one of the measurements did not follow a normal distribution.

To analyze the correlation between quantitative variables, Spearman's order correlation coefficient was calculated. The significance of the correlation coefficients was assessed by the correlation coefficient test whereby a coefficient is significantly non-zero if the *p*-value of the correlation test is less than the level of significance. As for the strength or intensity of the correlation, in this work, the correlation between two variables was considered strong enough only if the correlation coefficient had an absolute value greater than 0.7 and a moderate correlation was considered if it had an absolute value greater than 0.5 and less than or equal to 0.7.

All discussions about the significance tests were carried out considering a maximum significance level of 5% (0.05). Details of the proposed methodology of descriptive and inferential statistics can be found in Triola [65], Favero et al. [66], and Medronho et al. [67].

## 4. Results

Considering students' scholarly and learning performance during the COVID-19 pandemic period, Table 1 shows the percentage frequency distribution of the students who reduced, maintained, and improved their scholarly performance, according to the teacher's perception. Many aspects interfered in scholarly and learning performance during the period. Teachers' and students' feelings and sensations caused by social isolation and remoteness can be highlighted. Among the significant feelings raised in this research were strangeness and frustration. Lewis et al. [41] corroborate this idea, showing in their study that among the disadvantages arising from distance learning in the COVID-19 pandemic period are feelings of intimidation, confusion, and frustration. In addition, for Petrovic et al. [68], anxiety is the dominant feeling during the pandemic period.

**Table 1.** The students who reduced, maintained, and improved their scholarly performance.

| Students % | Reduced | | Maintained | | Improved | |
|---|---|---|---|---|---|---|
| | **N** | **%** | **N** | **%** | **N** | **%** |
| 0% | 8 | 7.2 | 10 | 9.1 | 46 | 42.2 |
| 0–20% | 10 | 9.0 | 27 | 24.5 | 38 | 34.9 |
| 20–40% | 19 | 17.1 | 29 | 26.4 | 15 | 13.8 |
| 40–60% | 19 | 17.1 | 17 | 15.5 | 5 | 4.6 |
| 60–80% | 22 | 19.8 | 17 | 15.5 | 2 | 1.8 |
| 80–100% | 28 | 25.2 | 7 | 6.4 | 2 | 1.8 |
| 100% | 5 | 4.5 | 3 | 2.7 | 1 | 0.9 |

The analysis presents evidence that the percentage of students who reduced their scholarly performance is typically in the range greater than or equal to 20% and less than 100% (79.2%), the percentage of students who maintained their scholarly performance is typically in the range between 0 and 40% (50.9%), and the percentage of students who improved their scholarly performance is typically in the range greater than or equal to 0% and less than 20% (77.1%).

Table 2 shows the percentage frequency distribution of students who reduced, maintained, and improved their learning performance, according to teacher perception. The percentage of students who reduced their learning performance is typically in the range greater than or equal to 40% and less than 100% (60.0%), the percentage of students who maintained their learning performance is typically in the range between 0 and 40% (75.6%), and the percentage of students who improved their learning performance is typically 0% (46.1%).

**Table 2.** The students who reduced, maintained, and improved their learning performance.

| Students % | Reduced | | Maintained | | Improved | |
|---|---|---|---|---|---|---|
| | N | % | N | % | N | % |
| 0% | 17 | 14.8 | 23 | 20.0 | 53 | 46.1 |
| 0–20% | 11 | 9.6 | 29 | 25.2 | 36 | 31.3 |
| 20–40% | 14 | 12.2 | 35 | 30.4 | 14 | 12.2 |
| 40–60% | 22 | 19.1 | 22 | 19.1 | 6 | 5.2 |
| 60–80% | 17 | 14.8 | 8 | 7.0 | 4 | 3.5 |
| 80–100% | 30 | 26.1 | 6 | 5.2 | 1 | 0.9 |
| 100% | 4 | 3.5 | 2 | 1.7 | 1 | 0.9 |

Table 3 shows the statistics of the main percentage of the students who reduced, maintained, and improved scholarly performance and reduced, maintained, and improved learning performance, according to the teachers' statement. By the values of the coefficients of variation, all greater than 0.4, it was possible to conclude that the teachers diverged greatly in their percentage declarations; that is, the distributions have high variability. In average terms, teachers declare that 53.2% of students reduced, 32.2% maintained, and only 11.3% improved their scholarly performance. In total, 52.9% of the students reduced, 29.9% maintained, and 13.4% improved their learning performance.

**Table 3.** The students who reduced, maintained, and improved their scholarly performance and the students who reduced, maintained, and improved their learning performance.

| | Reduced Scholarly Performance | Reduced Learning Performance | Maintained Scholarly Performance | Maintained Learning Performance | Improved Scholarly Performance | Improved Learning Performance |
|---|---|---|---|---|---|---|
| Minimum | 0.0 | 0.0 | 0.0 | 0.0 | 0.0 | 0.0 |
| Maximum | 100.0 | 100.0 | 100.0 | 100.0 | 100.0 | 100.0 |
| P25 | 30.0 | 25.0 | 7.5 | 10.0 | 0.0 | 0.0 |
| P50 | 50.0 | 50.0 | 20.0 | 20.0 | 5.0 | 5.0 |
| P75 | 80.0 | 80.0 | 52.5 | 50.0 | 15.0 | 20.0 |
| Average | 53.2 | 52.9 | 32.2 | 29.9 | 11.3 | 13.4 |
| Standard deviation | 31.5 | 31.8 | 28.8 | 26.5 | 19.4 | 20.5 |
| CV | 0.59 | 0.60 | 0.89 | 0.89 | 1.71 | 1.53 |
| *p*-value of the Wilcoxon test comparing the two distributions | 0.230 | | 0.130 | | 0.681 | |

P25 = 25th percentile, P50 = 50th percentile (median), P75 = 75th percentile, CV = coefficient of variation.

Table 4 provides statistics on the percentages of students who reduced, maintained, and improved scholarly performance and the students who reduced, maintained, and improved learning performance, according to the statement of teachers by the school's type.

While in the federal schools the percentage of students who reduced their scholarly performance was on average 28.2% and in the private schools, it was on average 38.5%, in the municipal and state schools, the percentages of students who reduced their scholarly performance were significantly higher, on average 56.8% in the municipal and 60.6% in the state schools. The discrepancy in the median terms is even greater: while in the federal school, the median percentage of students who reduced their scholarly performance was 13.5% and in the private schools, it was 30.0%, in the municipal and state schools, the median percentages of students who reduced their scholarly performance were significantly higher: 60.0% in the municipal and 60.5% in the state schools. While in the federal schools, the percentage of students who reduced their learning performance was on average 26.3%

and in the private schools, it was on average 28.2%, in the municipal and state schools, the percentages of students who reduced their learning performance were significantly higher, on average 50.4% in the municipal and 62.9% in the state schools. The discrepancy in the median terms is even greater: while in the federal schools, the median percentage of students who reduced their learning performance was only 3% and in the private schools, it was 20.0%, in the municipal and state schools, the median percentages of students who reduced their learning performance were significantly higher: 50.0% in the municipal schools and 70.0% in the state schools. Considering the Kruskal–Wallis test, there is no significant difference between the percentages of students from different schools that maintained their scholarly performance ($p$-value = 0.648) or between the percentages of students from different schools that maintained their learning ($p$-value = 0.929), nor between the percentages of students from different schools that improved their scholarly performance ($p$-value = 0.682) nor between the percentages of students from different schools that improved their learning ($p$-value = 0.516).

**Table 4.** The students who reduced, maintained, and improved their scholarly performance and the students who reduced, maintained, and improved their learning performance.

| School's Type | Reduced Scholarly Performance | | Reduced Learning Performance | | Maintained Scholarly Performance | | Maintained Learning Performance | | Improved Scholarly Performance | | Improved Learning Performance | |
|---|---|---|---|---|---|---|---|---|---|---|---|---|
| | $\underline{X}$ | $\widetilde{X}$ | $\underline{X}$ | $\widetilde{X}$ | $\underline{X}$ | $\widetilde{X}$ | $\underline{X}$ | $\widetilde{X}$ | $\underline{X}$ | $\widetilde{X}$ | $\underline{X}$ | $\widetilde{X}$ |
| Municipal School | 56.8 | 60.0 | 50.4 | 50.0 | 32.6 | 25.0 | 26.9 | 20.0 | 8.4 | 4.0 | 7.8 | 0.0 |
| State School | 60.6 | 60.5 | 62.9 | 70.0 | 26.9 | 20.0 | 23.0 | 20.0 | 7.9 | 1.0 | 11.9 | 5.0 |
| Federal School | 28.2 | 13.5 | 26.3 | 3.0 | 46.8 | 57.0 | 35.2 | 15.0 | 15.2 | 2.5 | 13.1 | 0.0 |
| Private School | 38.5 | 30.0 | 28.2 | 20.0 | 32.4 | 20.0 | 27.5 | 20.0 | 21.7 | 12.5 | 18.3 | 10.0 |
| $p$-value of the Kruskal–Wallis test comparing the distributions of different school types | 0.008 | | 0.001 | | 0.648 | | 0.929 | | 0.682 | | 0.516 | |

$\underline{X}$ : *average*, $\widetilde{X}$ : *median*.

Regarding the interaction's frequency distribution between the teachers and the students who reduced their scholarly and learning performance, for these students, the teachers declared several interactions, showing the commitment of the teachers to them. The most relevant interactions were to contact students through WhatsApp (12.9%), to contact students individually (12.1%), to encourage participation (10.3%), to be available for questions (6.9%), to ask students to redo their activities (5.2%), and to make recovery and revision possible (5.2%). For the interaction frequency distribution between the teachers and the students who maintained their scholarly and learning performance, the most relevant interactions were to encourage, to praise, and to incentivize (18.1%), to contact them through WhatsApp (12.9%), to contact them through the teaching platform (11.2%), to be available for questions (6.9%), to give individual attention (5.2%), and to apply digital methodologies such as choosing and sending good videos (5.2%). Considering the interaction frequency distribution between the teachers and the students who improved their scholarly or learning performance, the teachers declared several interactions and the most relevant interactions were to incentivize and encourage them, to praise, and to show happiness (19.0%), to increase the difficulty degree, and to present new and deeper content (6.9%), to contact them through email and teaching platform (6.0%), and to contact them through WhatsApp (5.2%).

Considering the themes of equality, quality, and justice in teaching methods during the COVID-19 pandemic period, the authors of this paper asked the teachers how they tried to establish them. The most relevant actions to establish equality, cited by the teachers, were to work with various digital media and different student access platforms (8.6%), to propose non-complex or playful activities, with interactive videos and simple language (8.6%), to produce explanatory videos (7.8%), to use more popular and easily accessible platforms such as WhatsApp and Facebook (7.8%), to make handouts, activities, and materials available online (6.0%), and to send motivational messages, to praise, and to incentivize (6.0%). Considering the actions declared by the teachers to establish quality in teaching and learning during the COVID-19 pandemic period, the most relevant actions, cited by the teachers, were to propose activities closer to the students' reality (11.3%), to search for dynamic and simple ways to teach classes (11.2%), to offer adequate service (10.3%), to be available to answer questions (10.3%), to produce materials rich in details and good quality handouts (10.3%), to post explanatory YouTube videos about the content worked on (9.5%), to explore the variety of media and digital resources (8.6%), to propose examples and challenges that stimulate the students (7.8%), to research on topics (6.0%), and to make adaptations in planning (5.2%). For the actions declared by teachers to establish justice in teaching and learning during the COVID-19 pandemic period, the most relevant actions, cited by the teachers, were to use popular platforms such as WhatsApp (6.9%), to abandon the meritocratic method (6.9%), to recognize and to understand students' reality with solidarity (6.9%), to share activities and different platforms and digital resources (6.0%), and to expand the variety of materials (6.0%).

The Kruskal–Wallis test showed a significant difference between the equality percentages achieved by teachers by schools' type ($p$-value = 0.010). It concluded that for teachers in federal schools, the percentage of the equality achieved was significantly higher than the percentage of the equality achieved in other school types. The post hoc paired tests showed that there was no significant difference between the equality percentages of the municipal, the state, and the private schools. Meanwhile, the test showed a significant difference between the percentages of quality achieved by the teachers from different school types ($p$-value = 0.007). The authors of this paper concluded that for the teachers in the federal schools and in the private schools, the percentages of quality achieved were significantly higher than the percentage of quality achieved in the municipal and in the state schools. The post hoc paired tests showed that there is no significant difference between the percentages of quality of the municipal and the state schools, and there is no significant difference between the quality percentages of the federal and the private schools. In addition, the test showed no significant difference between the percentages of justice achieved by teachers from the different school types ($p$-value = 0.120). So, the level of justice is not associated with the school type. Table 5 presents the teachers' declaration of equality (50.9%), quality (57.8%), and justice (58.7%).

**Table 5.** The main percentages of equality, quality, and justice.

|  | Equality | Quality | Justice |
|---|---|---|---|
| Minimum | 0.0% | 0.0% | 0.0% |
| Maximum | 100.0% | 100.0% | 100.0% |
| P25 | 25.0% | 41.5% | 30.0% |
| P50 | 50.0% | 60.0% | 63.0% |
| P75 | 75.0% | 80.0% | 90.0% |
| Average | 50.9% | 57.8% | 58.7% |
| StandardDeviation | 29.9% | 27.7% | 31.9% |
| CV | 0.59 | 0.48 | 0.54 |

P25 = 25th percentile, P50 = 50th percentile (median), P75 = 75th percentile, CV = coefficient of variation.

## 5. Discussion

Considering student scholarly and learning performance during the COVID-19 pandemic period, the rapid change in the scenario due to the pandemic did not allow the necessary adaption for the new educational reality. A great number of the students did not have the required home study environment, and this hindered their performance. Internet access and lack of equipment were the main teachers' and students' complaints during the COVID-19 pandemic period. The poor internet access made communication between teachers and students very difficult [46,52]. To reduce these inequalities, social programs for the provision of computers and internet access should be developed [3,4,46,52]. The design of learning activities is based on the principles of equity in health, supported by equity in access to education and learning for health. Cost and digital barriers often inhibit those who most need knowledge from accessing it [51].

Considering the type of schools in Brazil, as far as it is possible to see, there are differences in the management of federal, state, municipal, and private schools in the country and these differences can be reflected in student's scholarly and learning performance. The big challenge faced by teachers was to encourage students to continue their learning trajectory, minimizing the impact caused by these differences and by all the changes promoted by the COVID-19 pandemic period.

In all situations, the teachers tried to interact with their students. Many times, it was not possible due to the difficulties with internet access. Some students were unmotivated due to the changes imposed by the COVID-19 pandemic period. The students that benefited from good internet access created virtual classrooms so as not to lose contact with teachers and school friends. This helped them to overcome the COVID-19 pandemic period. It was concluded that the students with good internet access suffered less than those who did not have good internet access. Moreover, aspects such as teaching methods and principles, didactic techniques, and organizational forms can affect students' approach to the learning process [32,69,70].

The results show that all teachers, regardless of the school's type, tried to establish quality, equity, and justice for their students during the COVID-19 pandemic period and these actions' success was perceived. What is expected is that these actions to establish quality, equity, and justice for the students can be expanded to the post-COVID-19 pandemic period and that education continues to fulfill its role in guaranteeing full individual development, preparing for citizenship, and qualifying for the job market.

## 6. Conclusions and Limitations and Direction for Future Studies

The COVID-19 pandemic brought about many challenges and one of the greatest ones was related to education. Suddenly, students and teachers had to adapt to a new teaching modality and to the social isolation that interfered with their emotions and feelings. It brought about, despite new and positive learning experiences, some counterproductive consequences since it was suddenly imposed, without adequate time for those involved to properly assimilate the new methods. The percentage of students who improved their scholarly and learning performance was low when compared to the percentages of those who maintained or reduced their scholarly and learning performance. These results can be seen, in general, both in the results and in the concepts and careful observation of the teachers. However, teachers from private and federal schools demonstrated that there was a less marked reduction in student performance in these schools; that is, this negative result showed a higher percentage in municipal and state schools. All the teachers claimed to have sought to stimulate the learning of these students by seeking various other non-complex and playful activities, such as Facebook, WhatsApp, emails, and explanatory videos, and by making adaptations in the planning, trying to understand the reality of the students with solidarity, and expanding the variety of materials that were offered. Different and innovative technologies can be applied by teachers to motivate students [69,70], and this allows students to progress at their own individual pace [41]. In addition, the teachers sent motivational messages to offer quality, equality, and justice in teaching during the

COVID-19 pandemic period. In terms of equality, federal schools stood out more positively, with state and municipal schools presenting lower results in quality.

*Limitations and Directions for Further Research*

This study has limitations regarding the population. Due to the impossibility of interviewing the entire population, the authors of this paper developed a random sample of the universe. Another point of this exploratory research is the fact that it was developed in the city of Rio de Janeiro, which is in the state of Rio de Janeiro in the southeastern region of Brazil. Since Brazil is a continental country, the results cannot be generalized to all Brazilian cities and states.

In addition, this study is an exploratory one; there were no plans to check the hypotheses that were created. For future research, the expectation is to verify the directions taken by teaching activities during the COVID-19 pandemic period in other Brazilian regions, other countries, and in Brazilian and foreign universities. Another research agenda is to verify the effects of the post-COVID-19 pandemic in education and to expand the study on the inclusion of sustainable concepts in Brazilian schools and universities.

**Author Contributions:** Conceptualization, A.S. (Annibal Scavarda) and A.D.; methodology, A.D.; software, H.S.; validation, A.S. (Ana Scavarda), A.D. and A.R.; formal analysis, A.S. (Annibal Scavarda); investigation, A.D.; resources, A.S. (Annibal Scavarda); data curation, A.R.; writing—original draft preparation, A.D.; writing—review and editing, A.R.; visualization, H.S.; supervision, A.S. (Annibal Scavarda); project administration, A.S. (Annibal Scavarda); funding acquisition, H.S. All authors have read and agreed to the published version of the manuscript.

**Funding:** This study was financed in part by the Coordenação de Aperfeiçoamento de Pessoal de Nível Superior—Brasil (CAPES)—Finance Code 001 and in part by the Brazilian National Council for Scientific and Technological Development—CNPq [grant number 311881/2019-0].

**Institutional Review Board Statement:** The study was carried out in accordance with the parameters established by the ethics committee of the Federal University of Rio de Janeiro (Brazil) through the number 32959020.9.0000.5285. Opinion number: 4476475 (18 December 2020).

**Informed Consent Statement:** Informed consent was obtained from all subjects involved in the study.

**Data Availability Statement:** The data that support the findings of this study might be available from request.

**Acknowledgments:** We would like to express our gratitude to all the teachers and professors who, despite their busy schedules, collaborated with us and to all the professionals who worked hard to reduce the impacts of the COVID-19 pandemic.

**Conflicts of Interest:** The authors declare no conflict of interest.

## Appendix A

Multidisciplinary Research: Influence of Pandemics on Teaching and Learning
Email address
Name:
I authorize the results of this research to be presented and published, knowing that my name and my institution will be kept strictly confidential.
( ) Yes
( ) No
Sex:
Age:
Years as a teacher:
I am answering about the perspective of my performance before the pandemic in:
( ) Classroom Teaching
( ) E-Learning—EL
Type of school I teach (choose one):
( ) Municipal Public School

( ) State Public School
( ) Federal Public School
( ) Private school
( ) Other:
Class level I teach (choose one):
( ) Child education
( ) Elementary School I (First to Fifth Year)
( ) Elementary School II (Sixth to Ninth Year)
( ) High school
( ) Technological graduation
( ) Graduation
( ) Bachelor's degree
( ) Residence
( ) Specialization
( ) MBA
( ) Professional Master's
( ) Academic Master's
( ) Professional Doctorate (PHD)
( ) Doctorate (PHD) degree
( ) Post-doctoral
( ) Other:
The discipline that I teach and that I would like to use as a parameter for this research (write only one discipline). For example: Administration, Biology, Law, Economics, Physical Education, Nursing, Mechanical Engineering, Philosophy, Physics, History, Informatics, Mathematics, Music, Nutrition, Portuguese, Social Work, Theology, Tourism:

1-The difficulties and challenges I encounter in my teaching practice in the pandemic are (write "NO", if this does not apply):

2-I can overcome and adapt to these difficulties and challenges as follows (write "NO", if this does not apply):

3-The facilities and opportunities I find in my teaching practice in the pandemic are (write "NO", if this does not apply):

4-I can take advantage of these facilities and opportunities as follows (write "NO", if this does not apply):

5-The technology (s) and methodology (s) I use for my teaching practice in the pandemic are (you can choose more than one):
( ) Blackboard
( ) Email
( ) Facebook
( ) Google Classroom
( ) Google Meet
( ) Instagram
( ) Microsoft Teams
( ) Moodle
( ) Skype
( ) YouTube
( ) WhatsApp
( ) Zoom
( ) None
( ) Other:
6-The technology (s) and methodology (s) that I am creating familiarity with because of the pandemic are (you can choose more than one):
( ) Blackboard
( ) Email
( ) Facebook
( ) Instagram
( ) Google Classroom
( ) Google Meet

( ) Microsoft Teams
( ) Moodle
( ) Skype
( ) YouTube
( ) WhatsApp
( ) Zoom
( ) None
( ) Other:

7-The training (s) I am doing to work with this (these) technology (s) and these (these) methodology (s) are (write "NO", if this does not apply):

8-I consider myself apt to use this (these) technology (s) and this (these) methodology (s) (0% totally unfit and 100% totally fit):

9-The work from home and the consequent social isolation interfere with my work as follows (write "NO", if this does not apply):

10-I try to establish EQUALITY for and among my students in the pandemic as follows (write "NO", if this does not apply):

11-The percentage of EQUALITY that I am managing to establish is (write "NO", if this does not apply):

12-I try to establish QUALITY for and among my students in the pandemic as follows (write "NO", if this does not apply):

13-The percentage that I am managing to establish for QUALITY is (write "NO", if this does not apply):

14-I try to establish JUSTICE for and among my students in the pandemic as follows (write "NO", if this does not apply):

15-The percentage that I am managing to establish for JUSTICE is (write "NO", if this does not apply):

16-The sensations, feelings and emotions that describe my teaching experience in the pandemic are (you can choose more than one):

( ) Joy
( ) Love
( ) Calm
( ) Jealous
( ) Compassion
( ) Fault
( ) Despair
( ) Hope
( ) Strangeness
( ) Euphoria
( ) Excitement
( ) Happiness
( ) Frustration
( ) Gratitude
( ) Hostility
( ) Humor
( ) Indifference
( ) Fear
( ) Nostalgia
( ) Hate
( ) Dread
( ) Rage
( ) Satisfaction
( ) Surprise
( ) Boredom
( ) Nervous tension
( ) Sadness
( ) None
( ) Other:



17-The difficulties and challenges of my students with the new teaching methods practiced in the pandemic are (write "NO", if this does not apply):

18-The facilities and opportunities of my students with the new teaching methods practiced in the pandemic are (write "NO", if this does not apply):

19-The work from home and the consequent social isolation interfere with the performance and learning of my students as follows (write "NO", if this does not apply):

20-Defining student performance as the assessment of knowledge acquired in the classroom. The percentage of my students with a drop in school performance in the pandemic is:

21-The percentage of my students with MAINTENANCE of school performance in the pandemic is:

22-The percentage of my students with INCREASED school performance in the pandemic is:

23-I interact with my students with DECREASED school performance as follows (write "NO", if this does not apply):

24-I interact with my students with MAINTENANCE of school performance as follows (write "NO", if this does not apply):

25-I interact with my students with INCREASED school performance as follows (write "NO", if this does not apply):

26-Defining student learning as the process of behavior change obtained through experience built by emotional, neurological, relational, and environmental factors. The percentage of my students with DECREASED learning in the pandemic is:

27-The percentage of my students with MAINTENANCE of learning in the pandemic is:

28-The percentage of my students with INCREASED learning in the pandemic is:

29-I interact with my students with DECREASED learning as follows (write "NO", if this does not apply):

30-I interact with my students with MAINTENANCE of learning as follows (write "NO", if this does not apply):

31-I interact with my students with INCREASED learning as follows (write "NO", if this does not apply):

32-The percentage of my students with DECREASED learning who were familiar with this technology (s) and this methodology (s) adopted in the pandemic is (0% unfamiliar and 100% familiar):

33-The percentage of my MAINTENANCE learning students who were familiar with this (these) technology (s) and this (s) methodology (s) adopted in the pandemic is (0% unfamiliar and 100% familiar):

34-The percentage of my students with INCREASED learning who were familiar with this technology (s) and this methodology (s) adopted in the pandemic is (0% unfamiliar and 100% familiar):

35-The percentage of my students with DECREASED school performance who were familiar with this technology (s) and this methodology (s) adopted in the pandemic is (0% unfamiliar and 100% familiar):

36-The percentage of my students with MAINTENANCE of school performance who were familiar with this (these) technology (s) and this (these) methodology (s) adopted in the pandemic is (0% unfamiliar and 100% familiar):

37-The percentage of my students with INCREASED school performance who were familiar with this (these) technology (s) and this (these) methodology (s) adopted in the pandemic is (0% unfamiliar and 100% familiar):

38-The training that my students did to use the technology (s) and methodology (s) in the pandemic are (write "NO", if this does not apply):

39-My students' feelings and emotions that describe their learning experience in the pandemic are (you can choose more than one):

( ) Joy
( ) Love
( ) Calm
( ) Jealous
( ) Compassion
( ) Fault
( ) Despair
( ) Hope
( ) Strangeness
( ) Euphoria
( ) Excitement
( ) Happiness
( ) Frustration
( ) Gratitude
( ) Hostility
( ) Humor

( ) Indifference
( ) Fear
( ) Nostalgia
( ) Hate
( ) Dread
( ) Rage
( ) Satisfaction
( ) Surprise
( ) Boredom
( ) Tension
( ) Sadness
( ) None
( ) Other:

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
