# Peer review of "Equity, Justice, and Quality during the COVID-19 Pandemic Period: Considerations on Learning and Scholarly Performance in Brazilian Schools"

_education, doi:10.3390/educsci12050354_

Round 1

Reviewer 1 Report

Dear authors,

case study Equity, justice, and quality during the COVID-19 pandemical period: considerations on learning and scholar performance in the Brazilian schools is very interesting. Below are observations, comments and recommendations for its improvement.

Theoretical backround

I recommend adding studies from other parts of the world to the chapter as well.

Results

I propose to separate the results chapter and the discussion as separate chapters.

Discussion

I propose to adapt the discussion chapter according to the standards of scientific studies. The results need to be discussed and compared with other authors. I also recommend the literature on this issue. You can use these resources, for example:

Pavlíková, M.; Sirotkin, A.; Králik, R.; Petrikovičová, L.; Martin, J.G. How to Keep University Active during COVID-19 Pandemic: Experience from Slovakia. Sustainability 2021, 13, 10350. https://doi.org/10.3390/su131810350

Šolcová, L.; Magdin, M. Interactive textbook - A new tool in off-line and on-line education. Turkish Online Journal of Educational Technology. 2016, 15,(3)

Hašková, A.; Šafranko, C.; Pavlíková, M.; Petrikovičová, L. Application of online teaching tools and aids during corona pandemics. Ad Alta: Journal of Interdisciplinary Research, 2020, 10(2), 106-112,  https://doi.org/10.33543/1002.

Petrikovičová, L.; Dysková S.; Pavlíková M.; Vasbieva DG.; Kalugina OA. Teaching geographical methods and forms in the United States, Iceland and Slovakia. Ad Alta: Journal of Interdisciplinary Research, 2021, 11(1), 398-401, https://doi.org/10.33543/1101.

Petrikovičová, L.; Ďurinková, A.; Králik, R.; Kurilenko, V. Methodology of Working with a Textbook Versus Field Activities of Teaching Geography during the Corona Crisis.  European Journal of Contemporary Education, 2021, 10(2), 428-437,  https://doi.org/10.13187/ejced.2021.2.428.

Petrovič, F., Murgaš, F., & Králik, R. (2021). Happiness in Czechia during the COVID-19 Pandemic. Sustainability, 13(19), 10826. https://doi.org/10.3390/su131910826

Kobylarek, A.; Błaszczyński, K.; Ślósarz, L.; Madej, M.; Carmo, A.; Hlad, Ľ.; Králik, R.; Akimjak, A.; Judák, V.; Maturkanič, P.; Biryukova, Y.; Tokárová, B.; Martin, J.G.; Petrikovičová, L. The Quality of Life among University of the Third Age Students in Poland, Ukraine and Belarus. Sustainability 2022, 14, 2049. https://doi.org/10.3390/su14042049

References

This chapter is not written according to the requirements of the journal.

I wish you good luck in publishing your article.

Author Response

REVIEWER 1

Dear Reviewer,

We thank you very much for all your comments. Please, see below our answers to your questions and the suggested changes already done.

Kind regards,

The Authors

-------------------------------------------------------------------------------------------------------------------------------

Comments and Suggestions for Authors

The Reviewer 1 – Comment 1

case study Equity, justice, and quality during the COVID-19 pandemical period: considerations on learning and scholar performance in the Brazilian schools is very interesting. Below are observations, comments and recommendations for its improvement.

Comments of the authors about Reviewer 1 – Comment 1

Thank you very much for your observation. We believe that this subject is very relevant and needs to be widely discussed.

-------------------------------------------------------------------------------------------------------------------------------

The Reviewer 1 – Comment 2

Theoretical backround

I recommend adding studies from other parts of the world to the chapter as well.

Comments of the authors about Reviewer 1 – Comment 2

Thank you very much. New references were included.

Theoretical background

The theoretical section presents the theoretical research background. This paper examines the information of 61 papers achieved through scientific database searches that relate the themes COVID-19 and education.

2.1. The COVID-19 and the education

During the COVID-19 pandemical period, teachers and authorities made all the effort to provide the best experiences to the students. The idea was to overcome the social isolation [21, 27], all the unexpected situation, and to give adequate support over the COVID-19 pandemical period [28, 29, 30, 31, 32]. The social isolation has affected not only the students’ and professors’ health but also the institutions’ health [33, 34]. Despite of the difficulties faced during the COVID-19 pandemical period the schools and universities continued their work [32].

In this sense, the use of technologies in education was the solution found by the authorities to compensate for the lack of face-to-face education [35]. The strategy was well succeeded. The use of technology in education is close to become a new normal [3, 4, 36, 37, 38]. For Fischer et al. [38] learning and education should be rethought and reinvented through the integration with innovative technologies while Lacka et al. [39] believe that technologies are well integrated with education [40; 41; 42]. With online education the students can obtain a greater amount of information [40; 43; 44] and were able to build their knowledge base themselves [40].

One of the challenges encountered by the educational system during the COVID-19 pandemical period, was the fact that most students do not have the necessary resources to monitor the online activities [4, 45, 46]. Sallaberry et al. [47] highlighted that from the perspective of the professor during the COVID-19 pandemical period, the challenges are related to the reduction of time to develop the activities, and, in the perception of the student, the lack of motivation was outlined as one of the main challenges. The lack of an adequate infrastructure for some students, lack of teacher-student communication and interaction, lack of socialization, impossibility of performing practical applications and, lack of learning motivation are the main problems faced during the COVID-19 pandemical period [48].

Mok el al. [49] and Andrew et al. [50] affirmed that the COVID-19 pandemical crisis can intensify inequalities in educational system. Andrew et al. [51] concern about inequalities considering that low-income students have less possibility of accessing the internet and resources available at home like computers than rich students.

It was evident that the COVID-19 pandemic has affected the right to quality education and the process to guarantee social justice [52]. Studies shows a drop in quality when using online methods [53] while Gozzelino et al. [54] and Utunen et al. [55] affirm that is a big challenge to offer quality education during COVID-19 period. To improve quality education is recommended the improvement of the internet connection and the quality of the platforms, the education of students’ parents for the use of internet devices, and the free provision of equipment for low-income students. [56]. Also, the COVID-19 pandemical period was an opportunity to accelerate the construction of high-quality teachers [57].

Education has as main point to promote social justice and diversity [58]. Social justice is related to the possibility of people’s participation, especially the most vulnerable in the education process. Schools and universities must work for social justice, quality, and equity, thus becoming an important tool for social change [48]. Equity means that no one is left behind [59; 60].

For Dewart et al. [61] the COVID-19 pandemic will change forever the educational landscape and it will provide a great lesson in equity, leadership, social justice, and ethics. Torda [62] added that many changes that came with the COVID-19 pandemic will make better educators, collaborators, and innovators while Tracey and Tolan [63] affirmed that it is an opportunity to explore more effective teaching practices.

-------------------------------------------------------------------------------------------------------------------------------

The Reviewer 1 – Comment 3

Results

I propose to separate the results chapter and the discussion as separate chapters.

Comments of the authors about Reviewer 1 – Comment 3

Thank you very much. The results and discussion chapters were separated.

  1. Results

Considering student scholar and learning performance during the COVID-19 pandemical period, Table 1 shows the percentage frequency distribution of the students who reduced, maintained, and improved their scholar performance, according to the teacher's perception. Many aspects can interfere in scholar and learning performance during the period. Highlight can be given to the teachers’ and students’ feelings and sensations caused by social isolation and remoteness. Among the significant feelings raised in this research were strangeness and frustration. Lewis et al. [41] corroborate with this idea showing in their study that among the disadvantages from distance learning provoked by the COVID-19 pandemical period are feeling as intimidation, confusion, and frustration. Also, for Petrovic et al. [68] anxiety is the feeling is the dominant feeling during the pandemical period.

The analysis presents that the percentage of students who reduced their scholar performance is typically in the range greater than or equal to 20% and less than 100% (79.2%), the percentage of students who maintained their scholar performance is typically in the range between 0 and 40% (50.9%), and the percentage of students who improved their scholar performance is typically in the range greater than or equal to 0% and less than 20% (77.1%).

Table 1. The students who reduced, maintained, and improved their scholar performance.

Students %

Reduced

Maintained

Improved

N

%

N

%

N

%

0%

8

7.2

10

9.1

46

42.2

0% ⎯ 20%

10

9.0

27

24.5

38

34.9

20% |⎯ 40%

19

17.1

29

26.4

15

13.8

40% |⎯ 60%

19

17.1

17

15.5

5

4.6

60% |⎯ 80%

22

19.8

17

15.5

2

1.8

80% |⎯ 100%

28

25.2

7

6.4

2

1.8

100%

5

4.5

3

2.7

1

0.9

Table 2 shows the percentage frequency distribution of students who reduced, maintained, and improved their learning performance, according to teacher perception. The percentage of students who reduced their learning performance is typically in the range greater than or equal to 40% and less than 100% (60.0%), the percentage of students who maintained their learning performance is typically in the range between 0 and 40% (75.6%), and the percentage of students who improved their learning performance is typically 0% (46.1%).

Table 2. The students who reduced, maintained, and improved their learning performance.

Students %

Reduced

Maintained

Improved

N

%

N

%

N

%

0%

17

14.8

23

20.0

53

46.1

0% ⎯ 20%

11

9.6

29

25.2

36

31.3

20% |⎯ 40%

14

12.2

35

30.4

14

12.2

40% |⎯ 60%

22

19.1

22

19.1

6

5.2

60% |⎯ 80%

17

14.8

8

7.0

4

3.5

80% |⎯ 100%

30

26.1

6

5.2

1

0.9

100%

4

3.5

2

1.7

1

0.9

Table 3 shows the statistics of the main percentage of the students who reduced, maintained, and improved scholar performance; reduced, maintained, and improved learning performance, according to the teachers' statement. By the values ​​of the coefficients of variation, all greater than 0.4, it was possible to conclude that the teachers diverged a lot in their percentage declarations, that is, the distributions have high variability. In average terms, teachers declare that 53.2% of students reduced, 32.2% maintained and only 11.3% improved their scholar performance. For 52.9% of the students reduced, 29.9% maintained, and 13.4% of the students improved their learning performance.

Table 3. The students who reduced, maintained, and improved their scholar performance; and the students who reduced, maintained, and improved their learning performance.

Reduced

scholar performance

Reduced learning

Performance

Maintained scholar performance

Maintained learning

Performance

Improved scholar performance

Improved learning

Performance

Minimum

0.0

0.0

0.0

0.0

0.0

0.0

Maximum

100.0

100.0

100.0

100.0

100.0

100.0

P25

30.0

25.0

7.5

10.0

0.0

0.0

P50

50.0

50.0

20.0

20.0

5.0

5.0

P75

80.0

80.0

52.5

50.0

15.0

20.0

Average

53.2

52.9

32.2

29.9

11.3

13.4

Standard deviation

31.5

31.8

28.8

26.5

19.4

20.5

CV

0.59

0.60

0.89

0.89

1.71

1.53

p-value of the Wilcoxon test comparing the two distributions

0.230

0.130

0.681

P25= 25th percentile, P50= 50th percentile (median), P75= 75th percentile, CV = Coefficient of variation.

Table 4 provides statistics on the percentages of students who reduced, maintained, and improved scholar performance; and the students who reduced, maintained learning, and improved learning performance, according to the statement of teachers by the school’s type.

Table 4. The students who reduced, maintained, and improved their scholar performance; the students who reduced, maintained, and improved their learning performance.

School’s Type

Reduced

scholar performance

Reduced

learning performance

Maintained

scholar performance

Maintained

learning performance

Improved

scholar performance

Improved

learning performance

Municipal

School

56.8

60.0

50.4

50.0

32.6

25.0

26.9

20.0

8.4

4.0

7.8

0.0

State

School

60.6

60.5

62.9

70.0

26.9

20.0

23.0

20.0

7.9

1.0

11.9

5.0

Federal

School

28.2

13.5

26.3

3.0

46.8

57.0

35.2

15.0

15.2

2.5

13.1

0.0

Private

School

38.5

30.0

28.2

20.0

32.4

20.0

27.5

20.0

21.7

12.5

18.3

10.0

p-value of the Kruskal-Wallis test comparing the distributions of different schools’ types

0.008

0.001

0.648

0.929

0.682

0.516

While in the federal schools the percentage of students who reduced their scholar performance was on average 28.2% and in the private schools, it was on average 38.5%, in the municipal and state schools, the percentages of students who reduced their scholar performance were significantly higher, on average 56.8% in the municipal and 60.6% in the state schools. The discrepancy in the median terms is even greater: while in the federal school the median percentage of students who reduced their scholar performance was 13.5% and in the private schools, it was 30.0%, in the municipal and state schools the median percentages of students who reduced their scholar performance are significantly higher: 60.0% in the municipal and 60.5% in the state schools. While in the federal schools the percentage of students who reduced their learning performance was on average 26.3% and in the private schools, it was on average 28.2%, in the municipal and state schools the percentages of students who reduced their learning performance are significantly higher, on average 50.4% in the municipal and 62.9% in the state schools. The discrepancy in the median terms is even greater: while in the federal schools the median percentage of students who have reduced their learning performance was only 3% and in the private schools, it was 20.0%, in the municipal and state schools the median percentages of students who reduced their learning performance are significantly higher: 50.0% in the municipal schools and 70.0% in the state schools. Considering the Kruskal-Wallis test, there is no significant difference between the percentages of students from different schools that maintained their scholar performance (p-value = 0.648) or between the percentages of students from different schools that maintained their learning (p-value = 0.929), neither between the percentages of students from different schools that improved their scholar performance (p-value = 0.682) nor between the percentages of students from different schools that improved their learning (p-value = 0.516).

Regarding the interaction’s frequency distribution between the teachers and the students who reduced the scholar and learning performance, for these students, the teachers declared several interactions, showing the commitment of the teachers to them. The most relevant interactions were to contact through WhatsApp (12.9%), to contact individually contact (12.1%), to encourage the participation (10.3%), to be available for questions (6.9%), to ask them to redo the activities (5.2%), and to make recovery and revision (5.2%). For the interaction frequency distribution between the teachers and the students who maintained the scholar and learning performance, the most relevant were to encourage, to praise, and to incentive (18.1%), to contact them through WhatsApp (12.9%), to contact them through the teaching platform (11.2%), to be available for questions (6.9%), to give individual attention (5.2%), and to apply digital methodologies choosing and sending good videos (5.2%). Considering the interaction frequency distribution between the teachers and the student who improved the scholar or learning performance, the teachers declared several interactions and the most relevant interactions were to incentive and encourage them, to praise, and to show happiness (19.0%), to increase the difficulty degree and to present new and deeper content (6.9%), to contact them through email and teaching platform (6.0%), and to contact them through WhatsApp (5.2%).

Considering the themes of equality, quality, and justice in teaching methods during the COVID-19 pandemical period, the authors of this paper asked the teachers how they try to establish it. The most relevant actions to establish equality, cited by the teachers were to work with various digital media and different student access platforms (8.6%), to propose non-complex or playful activities, with interactive videos and simple languages (8.6%), to produce explanatory videos (7.8%), to use more popular and easily accessible platforms like WhatsApp and Facebook (7.8%), to leave handouts, activities, and materials available online (6.0 %), and to send motivational messages, to praise, and to incentive (6.0%). Considering the actions declared by the teachers to establish quality in teaching and learning during the COVID-19 pandemical period, the most relevant actions, cited by the teachers were to propose activities closer to the students' reality (11.3%), to search for dynamic and simple ways to teach classes (11.2%), to offer adequate service (10.3%), to be available to answer questions (10.3%), to produce materials rich in details and good quality handouts (10.3%), to post explanatory YouTube videos about the content worked on (9.5%), to explore the variety of media and digital resources (8.6%), to propose examples and challenges that stimulate the students (7.8%), to research on topics (6.0%), and to make adaptations in planning (5.2%). For the actions declared by teachers to establish justice in teaching and learning during the COVID-19 pandemical period, the most relevant actions, cited by the teachers, were to use popular platforms like WhatsApp (6.9%), to abandon the meritocratic method (6.9%), to know and to understand students' reality with solidarity (6.9%), to share activities and different platforms and digital resources (6.0%), and to expand the variety of materials (6.0%).

Table 5. The percentage frequency distribution of Equality, Quality, and Justice.

Percentage

              Equality

      Quality

   Justice

N

%

N

%

N

%

0%

3

3.5

4

4.0

3

3.6

0% 20%

8

9.3

8

8.0

10

11.9

20% |⎯ 40%

17

19.8

8

8.0

11

13.1

40% |⎯ 60%

23

26.7

26

26.0

14

16.7

60% |⎯ 80%

15

17.4

24

24.0

14

16.7

80% |⎯ 100%

14

16.3

26

26.0

19

22.6

100%

6

7.0

4

4.0

13

15.5

The Kruskal-Wallis test showed a significant difference between the equality percentages achieved by teachers by schools’ type (p-value = 0.010). It concluded that for teachers in federal schools, the percentage of the equality achieved was significantly higher than the percentage of the equality achieved in other school’s types. The post hoc paired tests showed that there was no significant difference between the equality percentages of the municipal, the state, and the private schools. Meanwhile, the test showed a significant difference between the percentages of quality achieved by the teachers from different school’s types (p-value = 0.007). The authors of this paper concluded that for the teachers in the federal schools and in the private schools, the percentages of quality achieved were significantly higher than the percentage of quality achieved in the municipal and in the state schools. The post hoc paired tests showed that there is no significant difference between the percentages of quality of the municipal and the state schools, and there is no significant difference between the quality percentages of the federal and the private schools. In addition, the test showed no significant difference between the percentages of justice achieved by teachers from the different school’s types (p-value = 0.120). So, the level of justice is not associated with the school’s type. Table 6 presents the teachers’ declaration of equality (50.9%), quality (57.8%), and justice (58.7%).

Table 6. The main percentages of equality, quality, and justice.

Equality

Quality

Justice

Minimum

0.0%

0.0%

0.0%

Maximum

100.0%

100.0%

100.0%

P25

25.0%

41.5%

30.0%

P50

50.0%

60.0%

63.0%

P75

75.0%

80.0%

90.0%

Average

50.9%

57.8%

58.7%

Standard

Deviation

29.9%

27.7%

31.9%

CV

0.59

0.48

0.54

P25= 25th percentile, P50= 50th percentile (median), P75= 75th percentile, CV = Coefficient of variation.

  1. Discussion

Considering student scholar and learning performance during the COVID-19 pandemical period, the rapid change in the scenario due to the pandemic did not allow the necessary adaption for the new educational reality. A great part of the students did not have the required home study environment and it hindered their performance. The internet access and lack of equipment were the main teachers’ and students’ complaints during the COVID-19 pandemical period. The poor internet made the communication between teachers and students very difficult [46; 52]. To reduce these inequalities social program to the provision of computers and internet access should be developed [3; 4; 46; 52]. The design of learning activities is based on the principles of equity to health, supported by equity in access to education, and learning for health. Cost and digital barriers often inhibit those who most need knowledge from accessing it [51]

Considering the type of school in Brazil, as far as it is possible to see, there are differences in the management of federal, state, municipal, and private schools in the country, and these differences can be reflected in student’s scholar and learning performance. The big challenge faced by teachers was to encourage students to continue their learning trajectory, minimizing the impact caused by these differences, and by all the changes promoted by the COVID-19 pandemical period.

In all situations, the teachers tried to interact with their students. Many times, it was not possible due to the difficulties with Internet access. Some students were unmotivated due to the changes imposed by the COVID-19 pandemical period. The students that were benefited from good Internet access have created virtual classrooms to not lose contact with teachers and school friends. This helped them to overcome the COVID-19 pandemical period. It was concluded that the students with good Internet access suffered less than those who did not have good Internet access. Also, aspects like teaching methods and principles, didactic and techniques organizational forms can affect students approach to learning process [32; 69; 70].

The results show that all teachers, regardless of the school’s type, tried to establish quality, equity, and justice for their students during the COVID-19 pandemical period and these actions’ success was perceived. What is expected is that these actions to establish quality, equity, and justice to the students can be expanded to the post-COVID-19 pandemical period and that the education continues to fulfill its role of guaranteeing the full individual development, preparing to citizenship exercise, and qualifying for the job market.

-------------------------------------------------------------------------------------------------------------------------------

The Reviewer 1 – Comment 4

Discussion

I propose to adapt the discussion chapter according to the standards of scientific studies. The results need to be discussed and compared with other authors. I also recommend the literature on this issue. You can use these resources, for example:

Pavlíková, M.; Sirotkin, A.; Králik, R.; Petrikovičová, L.; Martin, J.G. How to Keep University Active during COVID-19 Pandemic: Experience from Slovakia. Sustainability 2021, 13, 10350. https://doi.org/10.3390/su131810350

Šolcová, L.; Magdin, M. Interactive textbook - A new tool in off-line and on-line education. Turkish Online Journal of Educational Technology. 2016, 15,(3)

Hašková, A.; Šafranko, C.; Pavlíková, M.; Petrikovičová, L. Application of online teaching tools and aids during corona pandemics. Ad Alta: Journal of Interdisciplinary Research, 2020, 10(2), 106-112,  https://doi.org/10.33543/1002.

Petrikovičová, L.; Dysková S.; Pavlíková M.; Vasbieva DG.; Kalugina OA. Teaching geographical methods and forms in the United States, Iceland and Slovakia. Ad Alta: Journal of Interdisciplinary Research, 2021, 11(1), 398-401, https://doi.org/10.33543/1101.

Petrikovičová, L.; Ďurinková, A.; Králik, R.; Kurilenko, V. Methodology of Working with a Textbook Versus Field Activities of Teaching Geography during the Corona Crisis.  European Journal of Contemporary Education, 2021, 10(2), 428-437,  https://doi.org/10.13187/ejced.2021.2.428.

Petrovič, F., Murgaš, F., & Králik, R. (2021). Happiness in Czechia during the COVID-19 Pandemic. Sustainability, 13(19), 10826. https://doi.org/10.3390/su131910826

Kobylarek, A.; Błaszczyński, K.; Ślósarz, L.; Madej, M.; Carmo, A.; Hlad, Ľ.; Králik, R.; Akimjak, A.; Judák, V.; Maturkanič, P.; Biryukova, Y.; Tokárová, B.; Martin, J.G.; Petrikovičová, L. The Quality of Life among University of the Third Age Students in Poland, Ukraine and Belarus. Sustainability 2022, 14, 2049. https://doi.org/10.3390/su14042049

Comments of the authors about Reviewer 1 – Comment 4

Thank you very much. The references suggested were considered in our discussion.

-------------------------------------------------------------------------------------------------------------------------------

The Reviewer 1 – Comment 5

References

This chapter is not written according to the requirements of the journal.

Comments of the authors about Reviewer 1 – Comment 5

Thank you very much. The references were adjusted as can be seen below.

References

  1. Stanton: R., To, Q. G., Khalesi, S., Williams, S. L., Alley, S. J., Thwaite, T. L., Fenning, A. S., & Vandelanotte, C. (2020). “Depression, Anxiety and Stress during COVID-19: Associations with Changes in Physical Activity, Sleep, Tobacco and Alcohol Use in Australian Adults”. Int. J. Environ. Res. Public Health, Vol. 17, 4065.

  1. Dias, A., Scavarda, A., Reis, A., Silveira, H. & Ebecken, N. F. F. (2020). “Managerial Strategies for Long-Term Care Organization Professionals: COVID-19 Pandemic Impacts”. Sustainability, Vol.12, Number 22. https://doi.org/10.3390/su12229682.

  1. Scavarda, A., Dias, A., Reis, A., Silveira, H., & Santos, I. A. (2021). “COVID-19 Pandemic Sustainable Educational Innovation Management Proposal Framework”. Sustainability, Vol. 13, pp. 6391. https://doi.org/10.3390/su13116391.

  1. Dias, A., Scavarda, A., Silveira, H., Scavarda, L. F., Kondamareddy, K. K. (2021). “The Educational Online System: COVID-19 Demands, Trends, Implications, Challenges, Lessons, Insights, Opportunities, Outlooks, and Directions in the Work from Home”. Sustainability, Vol. 13, pp. 12197. https://doi.org/10.3390/su132112197.

  1. Kaden, U. (2020). “COVID-19 School Closure-Related Changes to the Professional Life of a K–12 Teacher”. Educ. Sci., Vol. 10, pp. 165. https://doi:10.3390/educsci10060165.

  1. Gandolfi, A. (2020). “Planning of school teaching during Covid-19”. Physica, Vol. 415, pp. 132753. https://doi.org/10.1016/j.physd.2020.132753.

  1. Kim, L. E., & Asbury, K. (2020). “Like a rug had been pulled from under you: The impact of COVID-19 on teachers in England during the first six weeks of the UK lockdown”. British Journal of Educational Psychology, Vol. 90, pp. 1062–1083. https://doi:10.1111/bjep.12381.

  1. Kim, C. J. H., & Padilla, A. M. (2020). “Technology for Educational Purposes Among Low-Income Latino Children Living in a Mobile Park in Silicon Valley: A Case Study Before and During COVID-19”. Hispanic Journal of Behavioral Sciences, Vol. 42, Number 4, pp. 497–514. https://doi: 10.1177/0739986320959764.

  1. Pulimeno, M., Piscitelli, P., Colazzo, S., Colao, A., & Miani, A. (2020). “Indoor air quality at school and students’ performance: Recommendations of the UNESCO Chair on Health Education and Sustainable Development & the Italian Society of Environmental Medicine (SIMA)”. Health Promotion Perspectives, Vol. 10, Number 3, pp. 169-174. https://doi: 10.34172/hpp.2020.29.

  1. Helmandollar, M. S. (2020). “Meeting Students Where They Are: Implementing Canvas for Successful Student Outreach”. Inquiry: The Journal of the Virginia Community Colleges, Vol. 23, Number 1. https://commons.vccs.edu/inquiry/vol23/iss1/14.

  1. Svalina, V., & Ivić, V. (2020). “Case study of a student with disabilities in a vocational school during the period of online virtual classes due to Covid-19”. World Journal of Education, Vol. 10, Number 4.  https://doi:10.5430/wje.v10n4p115.

  1. Holt, E. A., Heim, A. B., Tessens, E., & Walker, R. (2020). “Thanks for inviting me to the party: Virtual poster sessions as a way to connect in a time of disconnection”. Ecology and Evolution, Vol. 10, pp. 12423–12430. https://doi: 10.1002/ece3.6756.

  1. Aguilera-Hermida, A. P. (2020). “College students’ use and acceptance of emergency online learning due to COVID-19”. International Journal of Educational Research Open, Vol.1, Number 100011. http://doi: 1016/j.ijedro.2020.100011.

  1. Mishra, L., Gupta, T. & Shree, A. (2020). “Online teaching-learning in higher education during lockdown period of COVID-19 pandemic”. International Journal of Educational Research open, Vol. 1, Number 100012. https://doi.org/10.1016/j.ijedro.2020.100012.

  1. Harries, A.J., Lee, C., Jones, L. et al. (2021). “Effects of the COVID-19 pandemic on medical students: a multicenter quantitative study”. BMC Med Educ, Vol. 21, Number 14. https://doi.org/10.1186/s12909-020-02462-1

  1. Ali, U., Herbst, C. M., & Makridis, C. A. (2021). “The impact of COVID-19 on the U.S. childcare market: Evidence  from stay-at-home orders”. Economics of Education Review, Vol. 82, Number 102094.   https://doi.org/10.1016/j.econedurev.2021.102094.

  1. Code, J., Ralph, R., & Forde, K. (2020). “Pandemic designs for the future: perspectives of technology education teachers during COVID-19”. Information and Learning Sciences, Vol. 121, Number 5/6, pp. 419-431. https://doi 10.1108/ILS-04-2020-0112.

  1. Li, B.Z., Cao, N.W., Ren, C.X., Chu, X.J., Zhou, H.Y., & Guo, B. (2020). “Flipped classroom improves nursing students’ theoretical learning in China: A meta-analysis”. PLoS ONE, Vol. 15, Number 8, pp. https://doi.org/10.1371/journal.pone.0237926.

  1. Abuhammad, S. (2020). “Barriers to distance learning during the COVID-19 outbreak: A qualitative review from parents’ perspective”. Heliyon, Vol. 6, pp. e05482. https://doi.org/10.1016/j.heliyon.2020.e05482.

  1. Armstrong-Mensah, E., Ramsey-White, K., Yankey, B., & Self-Brown, S. (2020). “COVID-19 and Distance Learning: Effects on Georgia State University School of Public Health Students”. Front. Public Health, Vol. 8, pp. 576227. http://doi: 10.3389/fpubh.2020.576227.

  1. Abbasi, J. (2020). “Social Isolation the other COVID-19 threat in nursing homes”. JAMA, Vol. 324, Number 7.

  1. Tran, T., Hoang, A. D., Nguyen, T. T., Dinh, V. H., Nguyen, Y. C., & Pham, H. H. (2020). “Dataset of Vietnamese student’s learning habits during COVID-19”. Data in Brief, Vol. 30 pp. 105682. https://doi.org/10.1016/j.dib.2020.105682.

  1. Tran, T., Hoang, A.D., Nguyen, Y. C., Nguyen, L.C., Ta, N. T., Pham, Q. H., Pham, C. X., L, Q.A., Dinh, V. H., & Nguyen, T. T. (2020). “Toward Sustainable Learning during School Suspension: Socioeconomic, Occupational Aspirations, and Learning Behavior of Vietnamese Students during COVID-19”. Sustainability Vol. 12, pp. 4195. http://doi:10.3390/su12104195.

  1. Dias, A.C., & Reis, A.C. (2017). “Estágio Supervisionado em arquivologia: Pontos fortes e fracos e sugestões para de melhoria para o programa”. Ci. Inf., Vol. 46, pp. 84–105. https://doi.org 0.18225/ci.inf..v46i2.4145.

  1. Cutri, R. M., Mena, J., & Whiting, E. F. (2020). “Faculty readiness for online crisis teaching: transitioning to online teaching during the COVID-19 pandemic”. European Journal of Teacher Education, Vol. 43, Number 4, pp. 523–541. https://doi.org/10.1080/02619768.2020.1815702.
  2. Daú, G., Scavarda, A., Scavarda, L. F., & Portugal, V. J. T. (2019). “The Healthcare Sustainable Supply Chain 4.0: The Circular Economy Transition Conceptual Framework with the Corporate Social Responsibility Mirror”. Sustainability, Vol. 11, Number 18, pp. 5130. https://doi.org/10.3390/su11185130.

27.     Karademir, A., Yaman, F., & Saatçioğlu, Ö. (2020). “Challenges of higher education institutions against COVID-19: The case of Turkey”. Journal of Pedagogical Research, Vol. 4, Number 4, pp. 453-474.  http://dx.doi.org/10.33902/JPR.2020063574.

  1. Olena, O., Nguyen, T. K., & Balakrishnan, V. D. (2020). “International students in Australia – during and after COVID-19”. Higher Education Research & Development, Vol. 39, Number 7, pp. 1372-1376, DOI: 10.1080/07294360.2020.1825346.

  1. Almusharraf, N. M., & Khahro, S. H. (2020). “Students’ Satisfaction with Online Learning Experiences During the COVID-19 Pandemic”. International Journal of Emerging Technologies in Learning, Vol. 15, Number 21, 2020. https://doi.org/10.3991/ijet.v15i21.15647.

  1. König, J., Jäger-Biela, D. J., & Glutsch, N. (2020). “Adapting to online teaching during COVID-19 school closure: teacher education and teacher competence effects among early career teachers in Germany”, European Journal of Teacher Education, Vol. 43, Number 4, pp. 608-622, DOI: 10.1080/02619768.2020.1809650.

  1. Bolumole, M. (2020). “Student life in the age of COVID-19”, Higher Education Research & Development, Vol. 39, Number 7, pp. 1357-1361, DOI: 10.1080/07294360.2020.1825345.

  1. Pavlíková, M., Sirotkin, A., Králik, R., Petrikovičová, L., & Martin, J.G. How to Keep University Active during COVID-19 Pandemic: Experience from Slovakia. Sustainability 2021, 13, 10350. https://doi.org/10.3390/su131810350

  1. McGill, L. (2020). “Start-up company: how and why universities should nurture student friendships from day one”, Perspectives: Policy and Practice in Higher Education, Vol. 24, Number 1, pp. 4-7.

  1. Berwick, D. M. (2020). “COVID-19: Beyond Tomorrow: Choices for the “New Normal””, JAMA, Vol. 323 Number 21, pp. 2125-2126.

  1. Moorhouse, B. L. (2020). “Adaptations to a face-to-face initial teacher education course ‘forced’ online due to the COVID-19 pandemic”, Journal of Education for Teaching, Vol. 46, Number 4, pp. 609-611, DOI: 10.1080/02607476.2020.1755205.

  1. Rajhans, V., Memon, U., Patil, V., & Goyal, A. (2020). “Impact of COVID-19 on academic activities and way forward in Indian Optometry”. Journal of Optometry, Vol. 13, pp. 216-226.

  1. Lee, K., Fanguy, M., Lu, X. S., & Bligh, B. (2021). “Student learning during COVID-19: It was not as bad as we feared”, Distance Education, DOI: 10.1080/01587919.2020.1869529.

  1. Fischer, G., Lundin, J., & Lindberg, J. Ola. (2020). “Rethinking and reinventing learning, education and collaboration in the digital age—from creating technologies to transforming cultures”. The International Journal of Information and Learning Technology, Vol. 37, Number 5, pp. 241-252. DOI 10.1108/IJILT-04-2020-0051.

  1. Lacka, E., Wong, T. C., & Haddoud, M. Y. (2021). “Can digital technologies improve students’ efficiency? Exploring the role of Virtual Learning Environment and Social Media use in Higher Education”. Computers & Education, Vol. 163, pp. 104099. https://doi.org/10.1016/j.compedu.2020.104099.

  1. Šolcová, L., & Magdin, M. Interactive textbook - A new tool in off-line and on-line education. Turkish Online Journal of Educational Technology. 2016, 15 (3).

  1. Lewis, Peter, Osborne, Yvonne, Gray, Genevieve, & Lacaze, Anne-Marie. Design and delivery of a distance education programme: Educating Vietnamese nurse academics from Australia. Procedia: Social and Behavioral Sciences, 47, pp. 1462-1468. https://eprints.qut.edu.au/57259/

  1. Hašková, A., Šafranko, C., Pavlíková, M., & Petrikovičová, L. Application of online teaching tools and aids during corona pandemics. Ad Alta: Journal of Interdisciplinary Research, 2020, 10(2), 106-112,  https://doi.org/10.33543/1002.

  1. Kobylarek, A., Błaszczyński, K., Ślósarz, L., Madej, M., Carmo, A., Hlad, Ľ., Králik, R., Akimjak, A., Judák, V., Maturkanič, P., Biryukova, Y., Tokárová, B., Martin, J.G., & Petrikovičová, L. The Quality of Life among University of the Third Age Students in Poland, Ukraine and Belarus. Sustainability 2022, 14, 2049. https://doi.org/10.3390/su14042049.

44.     Tkac, L., & Schauer, F. Dissemination of science among students by remote experimentation. Cyprus International conference on educational research (CY-ICER-2012). Procedia - Social and Behavioral Sciences. 2012, 47, pp.1335-1340. DOI:10.1016/j.sbspro.2012.06.822.

45.       Diniz, V. L., & Silva, R. A. da. (2020). “Formação de professores no Período Pandêmico: (Im)Possibilidades de Ações e Acolhimento no Curso de Geografia da Uft/Araguaína”. Revista Docência do Ensino Superior, Vol. 10, pp. 1-18. DOI: https://doi.org/10.35699/2237-5864.2020.24711.

  1. Moser, K. M., Wei, T., & Brenner, D. (2021). “Remote teaching during COVID-19: Implications from a national survey of language educators”. System, Vol. 97, pp. 102431 https://doi.org/10.1016/j.system.2020.102431.

  1. Sallaberry, J. D. et al. (2020). “Desafios docentes em tempos de isolamento social: estudo com professores do curso de Ciências Contábeis”. Revista Docência do Ensino Superior, Vol. 10, Number e024774, pp. 1-22, 2020. https://doi.org/10.35699/2237-5864.2020.24774.

  1. Radu, Maria-Crina, Schnakovszky, C., Herghelegiu, E., Ciubotariu, Vlad-Andrei, & Cristea, I. The Impact of the COVID-19 Pandemic on the Quality of Educational Process: A Student Survey. Int. J. Environ. Res. Public Health 2020, 17, 7770; doi:10.3390/ijerph17217770.

  1. Mok, Ka Ho, Xiong, Weiyan, Ke, Guoguo, Cheung, & Joyce Oi Wun. (2021). “Impact of COVID-19 pandemic on international higher education and student mobility: Student perspectives from mainland China and Hong Kong”. International Journal of Educational Research, Vol. 105, Number 101718.  https://doi.org/10.1016/j.ijer.2020.101718.

  1. Andrew, A., Cattan, S., Dias, M. C., Farquharson, C., Kraftman, L., Krutikova, S., Phimister, A., & Sevilla, A. (2020). “Inequalities in Children’s Experiences of Home Learning during the COVID-19 Lockdown in England”. Fiscal Studies, Vol. 41, Number 3, pp. 653–683.

  1. Andrew, A., Cattan, S., Dias, M. C., Farquharson, C., Kraftman, L., Krutikova, S., Phimister, A.,  & Sevilla, A. Inequalities in Children’s Experiences of Home Learning during the COVID-19 Lockdown in England. FISCAL STUDIES, vol. 41, no. 3, pp. 653–683 (2020) 0143-5671.

  1. Jiménez Hernández, A.S., Cáceres-Muñoz, J., & Martín-Sánchez, M. Social Justice, Participation and School during the COVID-19—The International Project Gira por la Infancia. Sustainability 2021, 13, 2704. https://doi.org/10.3390/su13052704.

  1. Scheffers, F., Moonen, X., & Vugt, E. v. Assessing the quality of support and discovering sources of resilience during COVID-19 measures in people with intellectual disabilities by professional carers. Research in Developmental Disabilities 111 (2021) 103889. https://doi.org/10.1016/j.ridd.2021.103889.

  1. Gozzelino, G., & Matera, F. Pedagogical lines and critical consciousness for quality education at the time of the Covid-19 pandemic.  Form@re - Open Journal per la formazione in rete.  21, n. 3, pp. 191-199 2021. DOI: https://doi.org/10.36253/form-10178.

  1. Utunen, ,  Van Kerkhove, Maria D,  Tokar, A., O'Connell, G., Gamhewage, G., & Socé, I. One Year of Pandemic Learning Response: Benefits of Massive Online Delivery of the World Health Organization’s Technical Guidance. JMIR Public Health Surveill 2021 | vol. 7 | iss. 4 | e28945 | p. 3, https://publichealth.jmir.org/2021/4/e28945

  1. Miyah, Y., Benjelloun, M., Lairini, S., & Lahrichi, A.  COVID-19 Impact on Public Health, Environment, Human Psychology, Global Socioeconomy, and Education. Hindawi. Volume 2022, https://doi.org/10.1155/2022/5578284.

  1. Deng C, Yang S, Liu Q, Feng S, & Chen C (2021) Sustainable development and health assessment model of higher education in India: A mathematical modeling approach. PLoS ONE 16(12): e0261776. https://doi.org/10.1371/journal.pone.0261776

  1. Wipfli, H., & Withers, M. Engaging youth in global health and social justice: a decade of experience teaching a high school summer course, Global Health Action, 15:1, 1987045, 2022.  DOI: 10.1080/16549716.2021.1987045.

  1. Chan, I. L., Mowson, R., Alonso, J. P., Roberti, J., Contreras, M., & Velandia-González, M. Promoting immunization equity in Latin America and the Caribbean:Case studies, lessons learned, and their implication for COVID-19 vaccine Equity. Vaccine 40 (2022) 1977–1986. https://doi.org/10.1016/j.vaccine.2022.02.051.

  1. Baral, S, Chandler, R., Prieto, R. G., Gupta, S., Mishra, S., & Kulldorff, M. Leveraging epidemiological principles to evaluate Sweden’s COVID-19 Response. Annals of Epidemiology 54 (2021) 21e26.  https://doi.org/10.1016/j.annepidem.2020.11.005

  1.  Dewart, G., Corcoran, L., Thirsk, L., & Petrovic, K. “Nursing education in a pandemic: Academic challenges in response to COVID-19”. Nurse Education Today, Vol. 92, Number 104471. https://doi.org/10.1016/j.nedt.2020.104471.

  1. Torda, A. (2020). “How COVID-19 has pushed us into a medical education Revolution”. Internal Medicine Journal. Vol. 50, pp. 1150–1153. https://doi:10.1111/imj.14882.

  1. Tracey J. N. & Tolan, M. (2020). “Online Accounting Courses: Transition and Emerging Issues”. The CPA Journal. Vol. 90, Number 5, pp. 11.

  1. inf.br. (2018). Censo (2018). available at: http://www.escolas.inf.br/rj/rio-de-janeiro. (Accessed 14 August 2020).

  1. Triola, M. F. (2008). Introdução à Estatística. 10.a Edição. LTC- Rio de Janeiro.

  1. Favero, L. P., Belfiore, P., Silva, F. L., & Chan, B. L. (2009). Análise de dados: modelagem multivariada para tomada de decisões. Rio de Janeiro: Elsevier.

  1. Medronho, R. A., Bloch, K.V., Luiz, R. R., & Werneck, G. L. (2009).  São Paulo. Editora Atheneu.

  1. Petrovič, F., Murgaš, F., & Králik, R. (2021). Happiness in Czechia during the COVID-19 Pandemic. Sustainability, 13(19), 10826. https://doi.org/10.3390/su131910826.

  1. Petrikovičová, L., Dysková S., Pavlíková M., Vasbieva D.G., & Kalugina O. A. Teaching geographical methods and forms in the United States, Iceland and Slovakia. Ad Alta: Journal of Interdisciplinary Research, 2021, 11(1), 398-401, https://doi.org/10.33543/1101.

  1. Petrikovičová, L., Ďurinková, A., Králik, R., & Kurilenko, V. Methodology of Working with a Textbook Versus Field Activities of Teaching Geography during the Corona Crisis.  European Journal of Contemporary Education, 2021, 10(2), 428-437,  https://doi.org/10.13187/ejced.2021.2.428.

-------------------------------------------------------------------------------------------------------------------------------

The Reviewer 1 – Comment 6

I wish you good luck in publishing your article.

Comments of the authors about Reviewer 1 – Comment 6

Thank you very much. We hope this article can be published in this respect journal.

Reviewer 2 Report

TITLE: The title does not express what the article is about.

ABSTRACT: In the abstract there is no need to mention the computer package used to analyse the data. Neither it is necessary to mention the statistical tests applied. The methodology used and the sample from which the data were obtained should be explained. The summary does not give a clear idea of what is explained in the article.

INTRODUCTION. It states that there are no studies (research) on children's learning habits during COVID-19, but then refers to several studies. This statement needs to be revised.

THEORETICAL FRAMEWORK. It is badly ordered, before explaining the part of materials and method it is important to make a review of the scientific literature, following the scientific method. Section 3 is badly placed in the article,

The review is very synthetic and imprecise. Much has been published in the last year on these issues. It should go into more depth and especially in relation to what is being studied: academic performance, equity, social justice.

There is no definition of the difference between academic performance and learning achievement and these are issues that are being studied.

There is talk of different typologies of schools, but they are not explained and this confuses the reader: why were these typologies of schools selected? was the selection proportional to the population distribution? 

There is talk of equality, quality, justice, but no clear definition of these concepts is given. What does equality imply? For example: economic, social, gender, cultural equality?

METHODS. The sample is large, teachers are surveyed online. It is not explained how the sampling was carried out and if the informants are representative according to the distribution of the typologies of schools present in the population.

The research questions are missing and the objective is very generic.

The research methodology is not clearly specified, although the statistical techniques to be used are exhaustively expressed. The characteristics of the questionnaire, whether it was based on closed questions, had a Likert scale, etc., are not explained.

RESULTS. The data are presented in a disordered way.  For example: first the percentage frequency distribution of students who reduced, maintained and improved their school performance is explained. Then they explain possible factors that influenced, as perceived by the teachers, the performance data, and then the previous percentage frequency distribution is explained again.

I advise to distribute the presentation of the results according to the blocks that are analysed.

The authors point out that considering the Kruskal-Wallis test, there are no significant differences between the percentages of students from different schools, however part of the discourse points out percentages to visualise possible differences. All this needs to be explained further.

DISCUSSION. There is no discussion of results in the light of scientific knowledge.

CONCLUSIONS: They are synthetic and reiterate the results presented. They should be redone.

 BIBLIOGRAPHICAL REFERENCES. Very few and it would be necessary to incorporate more references on the subject studied that have been generated in recent years.

Author Response

REVIEWER 2

Dear Reviewer,

We thank you very much for all your comments. Please, see below our answers to your questions and the suggested changes already done.

Kind regards,

The Authors

-------------------------------------------------------------------------------------------------------------------------------

The Reviewer 2 – Comment 1

ABSTRACT: In the abstract there is no need to mention the computer package used to analyse the data. Neither it is necessary to mention the statistical tests applied. The methodology used and the sample from which the data were obtained should be explained. The summary does not give a clear idea of what is explained in the article.

Comments of the authors about Reviewer 2 – Comment 1

Thank you very much. The abstract was adjusted as can be seen below:

Abstract: Due to the imperative need for change in habits caused by the COVID-19 pandemic that has plagued the world, this exploratory study plans to analyze the directions taken by teaching activities in public and private schools of the city of Rio de Janeiro (Brazil) and their consequences on learning and scholar performance concerning elementary and middle schools. In this way, this study verifies through an email questionnaire if there were equality, justice, and quality in teaching methods during the COVID-19 pandemic. This study had as a population the total number of K-12 teachers in the city of Rio de Janeiro. It was based on the 2018 Schools Census (escolas.inf.br, 2018) that registered 66,999 teachers in the city distributed in 2010 private schools, 1,439 municipal schools, 457 state schools, and 28 federal schools. To obtain valid results and due to the impossibility of interviewing the entire population, the authors of this paper developed a random sample of the universe. The results show that teachers tried to interact with students to overcome the problems faced during the COVID-19 pandemic period. Additionally, the study showed that there were differences in scholar and learning performance, equality, and quality in the types of schools analyzed. This paper will help to fill the literature gap on the subject and will boost ongoing discussion on the inclusion of sustainable concepts in education.

 -------------------------------------------------------------------------------------------------------------------------------

The Reviewer 2 – Comment

INTRODUCTION. It states that there are no studies (research) on children's learning habits during COVID-19, but then refers to several studies. This statement needs to be revised.

Comments of the authors about Reviewer 2 – Comment 2

Thank you very much. The introduction was adjusted as can be seen below:

  1. Introduction

The COVID-19 (SARS-CoV-2 Coronavirus) pandemic imposed dramatic changes to daily human activities [1, 2] and one of the biggest changes happened in education [3, 4]. School closures [5, 6. 7, 8, 9] after the COVID-19 appearance compelled schools to transition to online programs [10, 11, 12, 13, 14]. For Harries et al. [15] the COVID-19 pandemic has affected all level of education from the preschool to the postgraduation. All the effort were made to mitigate the spread of the COVID-19 pandemic [16].

Educators all over the world had to find ways of supporting the needs of their students in an online system [17] and they had to be technically prepared for this endeavor. For Li et al. [18], the teaching method directly impacts the students’ learning performance. Also, providing online resources became another way of overcoming distance learning barriers [19, 20, 21]. Children's learning habits during the COVID-19 period needs to be deeply researched [3, 4, 22, 23, 24].

The authors affirm that students’ learning habits differ according to social class, aspiration, and educational degree. The combination between affection, humbleness, optimism, and empathy allows mitigating online teaching mistakes and risks [25, 26]. This exploratory study has as main goal to observe students’ scholar and learning performance during the COVID-19 pandemic in public and private schools of the city of Rio de Janeiro (Brazil) with respect to elementary and middle schools. It will verify through an email questionnaire if there were equality, justice, and quality in teaching methods during the COVID-19 pandemic. This paper is divided into 5 Sections including this introductory one. Section 2 describes the theoretical background, Section 3 presents the methods, Section 4 provides the results and discussion, and Section 5 presents the conclusion.

-------------------------------------------------------------------------------------------------------------------------------

The Reviewer 2 – Comment 3

THEORETICAL FRAMEWORK. It is badly ordered, before explaining the part of materials and method it is important to make a review of the scientific literature, following the scientific method. Section 3 is badly placed in the article,

Comments of the authors about Reviewer 2 – Comment 3

Thank you very much. The theoretical framework was adjusted as can be seen below:

  1. Theoretical background

The theoretical section presents the theoretical research background. This paper examines the information of 61 papers achieved through scientific database searches that relate the themes COVID-19 and education.

2.1. The COVID-19 and the education

During the COVID-19 pandemical period, teachers and authorities made all the effort to provide the best experiences to the students. The idea was to overcome the social isolation [21, 27], all the unexpected situation, and to give adequate support over the COVID-19 pandemical period [28, 29, 30, 31, 32]. The social isolation has affected not only the students’ and professors’ health but also the institutions’ health [33, 34]. Despite of the difficulties faced during the COVID-19 pandemical period the schools and universities continued their work [32].

In this sense, the use of technologies in education was the solution found by the authorities to compensate for the lack of face-to-face education [35]. The strategy was well succeeded. The use of technology in education is close to become a new normal [3, 4, 36, 37, 38]. For Fischer et al. [38] learning and education should be rethought and reinvented through the integration with innovative technologies while Lacka et al. [39] believe that technologies are well integrated with education [40; 41; 42]. With online education the students can obtain a greater amount of information [40; 43; 44] and were able to build their knowledge base themselves [40].

One of the challenges encountered by the educational system during the COVID-19 pandemical period, was the fact that most students do not have the necessary resources to monitor the online activities [4, 45, 46]. Sallaberry et al. [47] highlighted that from the perspective of the professor during the COVID-19 pandemical period, the challenges are related to the reduction of time to develop the activities, and, in the perception of the student, the lack of motivation was outlined as one of the main challenges. The lack of an adequate infrastructure for some students, lack of teacher-student communication and interaction, lack of socialization, impossibility of performing practical applications and, lack of learning motivation are the main problems faced during the COVID-19 pandemical period [48].

Mok el al. [49] and Andrew et al. [50] affirmed that the COVID-19 pandemical crisis can intensify inequalities in educational system. Andrew et al. [51] concern about inequalities considering that low-income students have less possibility of accessing the internet and resources available at home like computers than rich students.

It was evident that the COVID-19 pandemic has affected the right to quality education and the process to guarantee social justice [52]. Studies shows a drop in quality when using online methods [53] while Gozzelino et al. [54] and Utunen et al. [55] affirm that is a big challenge to offer quality education during COVID-19 period. To improve quality education is recommended the improvement of the internet connection and the quality of the platforms, the education of students’ parents for the use of internet devices, and the free provision of equipment for low-income students. [56]. Also, the COVID-19 pandemical period was an opportunity to accelerate the construction of high-quality teachers [57].

Education has as main point to promote social justice and diversity [58]. Social justice is related to the possibility of people’s participation, especially the most vulnerable in the education process. Schools and universities must work for social justice, quality, and equity, thus becoming an important tool for social change [48]. Equity means that no one is left behind [59; 60].

For Dewart et al. [61] the COVID-19 pandemic will change forever the educational landscape and it will provide a great lesson in equity, leadership, social justice, and ethics. Torda [62] added that many changes that came with the COVID-19 pandemic will make better educators, collaborators, and innovators while Tracey and Tolan [63] affirmed that it is an opportunity to explore more effective teaching practices.

-------------------------------------------------------------------------------------------------------------------------------

The Reviewer 2 – Comment 4

 RESULTS. The data are presented in a disordered way.  For example: first the percentage frequency distribution of students who reduced, maintained ,and improved their school performance is explained. Then they explain possible factors that influenced, as perceived by the teachers, the performance data, and then the previous percentage frequency distribution is explained again.

DISCUSSION. There is no discussion of results in the light of scientific knowledge.

Comments of the authors about Reviewer 2 – Comment 4

  1. Results

Considering student scholar and learning performance during the COVID-19 pandemical period, Table 1 shows the percentage frequency distribution of the students who reduced, maintained, and improved their scholar performance, according to the teacher's perception. Many aspects can interfere in scholar and learning performance during the period. Highlight can be given to the teachers’ and students’ feelings and sensations caused by social isolation and remoteness. Among the significant feelings raised in this research were strangeness and frustration. Lewis et al. [41] corroborate with this idea showing in their study that among the disadvantages from distance learning provoked by the COVID-19 pandemical period are feeling as intimidation, confusion, and frustration. Also, for Petrovic et al. [68] anxiety is the feeling is the dominant feeling during the pandemical period.

The analysis presents that the percentage of students who reduced their scholar performance is typically in the range greater than or equal to 20% and less than 100% (79.2%), the percentage of students who maintained their scholar performance is typically in the range between 0 and 40% (50.9%), and the percentage of students who improved their scholar performance is typically in the range greater than or equal to 0% and less than 20% (77.1%).

Table 1. The students who reduced, maintained, and improved their scholar performance.

Students %

Reduced

Maintained

Improved

N

%

N

%

N

%

0%

8

7.2

10

9.1

46

42.2

0% ⎯ 20%

10

9.0

27

24.5

38

34.9

20% |⎯ 40%

19

17.1

29

26.4

15

13.8

40% |⎯ 60%

19

17.1

17

15.5

5

4.6

60% |⎯ 80%

22

19.8

17

15.5

2

1.8

80% |⎯ 100%

28

25.2

7

6.4

2

1.8

100%

5

4.5

3

2.7

1

0.9

Table 2 shows the percentage frequency distribution of students who reduced, maintained, and improved their learning performance, according to teacher perception. The percentage of students who reduced their learning performance is typically in the range greater than or equal to 40% and less than 100% (60.0%), the percentage of students who maintained their learning performance is typically in the range between 0 and 40% (75.6%), and the percentage of students who improved their learning performance is typically 0% (46.1%).

Table 2. The students who reduced, maintained, and improved their learning performance.

Students %

Reduced

Maintained

Improved

N

%

N

%

N

%

0%

17

14.8

23

20.0

53

46.1

0% ⎯ 20%

11

9.6

29

25.2

36

31.3

20% |⎯ 40%

14

12.2

35

30.4

14

12.2

40% |⎯ 60%

22

19.1

22

19.1

6

5.2

60% |⎯ 80%

17

14.8

8

7.0

4

3.5

80% |⎯ 100%

30

26.1

6

5.2

1

0.9

100%

4

3.5

2

1.7

1

0.9

Table 3 shows the statistics of the main percentage of the students who reduced, maintained, and improved scholar performance; reduced, maintained, and improved learning performance, according to the teachers' statement. By the values ​​of the coefficients of variation, all greater than 0.4, it was possible to conclude that the teachers diverged a lot in their percentage declarations, that is, the distributions have high variability. In average terms, teachers declare that 53.2% of students reduced, 32.2% maintained and only 11.3% improved their scholar performance. For 52.9% of the students reduced, 29.9% maintained, and 13.4% of the students improved their learning performance.

Table 3. The students who reduced, maintained, and improved their scholar performance; and the students who reduced, maintained, and improved their learning performance.

Reduced

scholar performance

Reduced learning

Performance

Maintained scholar performance

Maintained learning

Performance

Improved scholar performance

Improved learning

Performance

Minimum

0.0

0.0

0.0

0.0

0.0

0.0

Maximum

100.0

100.0

100.0

100.0

100.0

100.0

P25

30.0

25.0

7.5

10.0

0.0

0.0

P50

50.0

50.0

20.0

20.0

5.0

5.0

P75

80.0

80.0

52.5

50.0

15.0

20.0

Average

53.2

52.9

32.2

29.9

11.3

13.4

Standard deviation

31.5

31.8

28.8

26.5

19.4

20.5

CV

0.59

0.60

0.89

0.89

1.71

1.53

p-value of the Wilcoxon test comparing the two distributions

0.230

0.130

0.681

P25= 25th percentile, P50= 50th percentile (median), P75= 75th percentile, CV = Coefficient of variation.

Table 4 provides statistics on the percentages of students who reduced, maintained, and improved scholar performance; and the students who reduced, maintained learning, and improved learning performance, according to the statement of teachers by the school’s type.

Table 4. The students who reduced, maintained, and improved their scholar performance; the students who reduced, maintained, and improved their learning performance.

School’s Type

Reduced

scholar performance

Reduced

learning performance

Maintained

scholar performance

Maintained

learning performance

Improved

scholar performance

Improved

learning performance

Municipal

School

56.8

60.0

50.4

50.0

32.6

25.0

26.9

20.0

8.4

4.0

7.8

0.0

State

School

60.6

60.5

62.9

70.0

26.9

20.0

23.0

20.0

7.9

1.0

11.9

5.0

Federal

School

28.2

13.5

26.3

3.0

46.8

57.0

35.2

15.0

15.2

2.5

13.1

0.0

Private

School

38.5

30.0

28.2

20.0

32.4

20.0

27.5

20.0

21.7

12.5

18.3

10.0

p-value of the Kruskal-Wallis test comparing the distributions of different schools’ types

0.008

0.001

0.648

0.929

0.682

0.516

While in the federal schools the percentage of students who reduced their scholar performance was on average 28.2% and in the private schools, it was on average 38.5%, in the municipal and state schools, the percentages of students who reduced their scholar performance were significantly higher, on average 56.8% in the municipal and 60.6% in the state schools. The discrepancy in the median terms is even greater: while in the federal school the median percentage of students who reduced their scholar performance was 13.5% and in the private schools, it was 30.0%, in the municipal and state schools the median percentages of students who reduced their scholar performance are significantly higher: 60.0% in the municipal and 60.5% in the state schools. While in the federal schools the percentage of students who reduced their learning performance was on average 26.3% and in the private schools, it was on average 28.2%, in the municipal and state schools the percentages of students who reduced their learning performance are significantly higher, on average 50.4% in the municipal and 62.9% in the state schools. The discrepancy in the median terms is even greater: while in the federal schools the median percentage of students who have reduced their learning performance was only 3% and in the private schools, it was 20.0%, in the municipal and state schools the median percentages of students who reduced their learning performance are significantly higher: 50.0% in the municipal schools and 70.0% in the state schools. Considering the Kruskal-Wallis test, there is no significant difference between the percentages of students from different schools that maintained their scholar performance (p-value = 0.648) or between the percentages of students from different schools that maintained their learning (p-value = 0.929), neither between the percentages of students from different schools that improved their scholar performance (p-value = 0.682) nor between the percentages of students from different schools that improved their learning (p-value = 0.516).

Regarding the interaction’s frequency distribution between the teachers and the students who reduced the scholar and learning performance, for these students, the teachers declared several interactions, showing the commitment of the teachers to them. The most relevant interactions were to contact through WhatsApp (12.9%), to contact individually contact (12.1%), to encourage the participation (10.3%), to be available for questions (6.9%), to ask them to redo the activities (5.2%), and to make recovery and revision (5.2%). For the interaction frequency distribution between the teachers and the students who maintained the scholar and learning performance, the most relevant were to encourage, to praise, and to incentive (18.1%), to contact them through WhatsApp (12.9%), to contact them through the teaching platform (11.2%), to be available for questions (6.9%), to give individual attention (5.2%), and to apply digital methodologies choosing and sending good videos (5.2%). Considering the interaction frequency distribution between the teachers and the student who improved the scholar or learning performance, the teachers declared several interactions and the most relevant interactions were to incentive and encourage them, to praise, and to show happiness (19.0%), to increase the difficulty degree and to present new and deeper content (6.9%), to contact them through email and teaching platform (6.0%), and to contact them through WhatsApp (5.2%).

Considering the themes of equality, quality, and justice in teaching methods during the COVID-19 pandemical period, the authors of this paper asked the teachers how they try to establish it. The most relevant actions to establish equality, cited by the teachers were to work with various digital media and different student access platforms (8.6%), to propose non-complex or playful activities, with interactive videos and simple languages (8.6%), to produce explanatory videos (7.8%), to use more popular and easily accessible platforms like WhatsApp and Facebook (7.8%), to leave handouts, activities, and materials available online (6.0 %), and to send motivational messages, to praise, and to incentive (6.0%). Considering the actions declared by the teachers to establish quality in teaching and learning during the COVID-19 pandemical period, the most relevant actions, cited by the teachers were to propose activities closer to the students' reality (11.3%), to search for dynamic and simple ways to teach classes (11.2%), to offer adequate service (10.3%), to be available to answer questions (10.3%), to produce materials rich in details and good quality handouts (10.3%), to post explanatory YouTube videos about the content worked on (9.5%), to explore the variety of media and digital resources (8.6%), to propose examples and challenges that stimulate the students (7.8%), to research on topics (6.0%), and to make adaptations in planning (5.2%). For the actions declared by teachers to establish justice in teaching and learning during the COVID-19 pandemical period, the most relevant actions, cited by the teachers, were to use popular platforms like WhatsApp (6.9%), to abandon the meritocratic method (6.9%), to know and to understand students' reality with solidarity (6.9%), to share activities and different platforms and digital resources (6.0%), and to expand the variety of materials (6.0%).

Table 5. The percentage frequency distribution of Equality, Quality, and Justice.

Percentage

              Equality

      Quality

   Justice

N

%

N

%

N

%

0%

3

3.5

4

4.0

3

3.6

0% 20%

8

9.3

8

8.0

10

11.9

20% |⎯ 40%

17

19.8

8

8.0

11

13.1

40% |⎯ 60%

23

26.7

26

26.0

14

16.7

60% |⎯ 80%

15

17.4

24

24.0

14

16.7

80% |⎯ 100%

14

16.3

26

26.0

19

22.6

100%

6

7.0

4

4.0

13

15.5

The Kruskal-Wallis test showed a significant difference between the equality percentages achieved by teachers by schools’ type (p-value = 0.010). It concluded that for teachers in federal schools, the percentage of the equality achieved was significantly higher than the percentage of the equality achieved in other school’s types. The post hoc paired tests showed that there was no significant difference between the equality percentages of the municipal, the state, and the private schools. Meanwhile, the test showed a significant difference between the percentages of quality achieved by the teachers from different school’s types (p-value = 0.007). The authors of this paper concluded that for the teachers in the federal schools and in the private schools, the percentages of quality achieved were significantly higher than the percentage of quality achieved in the municipal and in the state schools. The post hoc paired tests showed that there is no significant difference between the percentages of quality of the municipal and the state schools, and there is no significant difference between the quality percentages of the federal and the private schools. In addition, the test showed no significant difference between the percentages of justice achieved by teachers from the different school’s types (p-value = 0.120). So, the level of justice is not associated with the school’s type. Table 6 presents the teachers’ declaration of equality (50.9%), quality (57.8%), and justice (58.7%).

Table 6. The main percentages of equality, quality, and justice.

Equality

Quality

Justice

Minimum

0.0%

0.0%

0.0%

Maximum

100.0%

100.0%

100.0%

P25

25.0%

41.5%

30.0%

P50

50.0%

60.0%

63.0%

P75

75.0%

80.0%

90.0%

Average

50.9%

57.8%

58.7%

Standard

Deviation

29.9%

27.7%

31.9%

CV

0.59

0.48

0.54

P25= 25th percentile, P50= 50th percentile (median), P75= 75th percentile, CV = Coefficient of variation.

  1. Discussion

Considering student scholar and learning performance during the COVID-19 pandemical period, the rapid change in the scenario due to the pandemic did not allow the necessary adaption for the new educational reality. A great part of the students did not have the required home study environment and it hindered their performance. The internet access and lack of equipment were the main teachers’ and students’ complaints during the COVID-19 pandemical period. The poor internet made the communication between teachers and students very difficult [46; 52]. To reduce these inequalities social program to the provision of computers and internet access should be developed [3; 4; 46; 52]. The design of learning activities is based on the principles of equity to health, supported by equity in access to education, and learning for health. Cost and digital barriers often inhibit those who most need knowledge from accessing it [51]

Considering the type of school in Brazil, as far as it is possible to see, there are differences in the management of federal, state, municipal, and private schools in the country, and these differences can be reflected in student’s scholar and learning performance. The big challenge faced by teachers was to encourage students to continue their learning trajectory, minimizing the impact caused by these differences, and by all the changes promoted by the COVID-19 pandemical period.

In all situations, the teachers tried to interact with their students. Many times, it was not possible due to the difficulties with Internet access. Some students were unmotivated due to the changes imposed by the COVID-19 pandemical period. The students that were benefited from good Internet access have created virtual classrooms to not lose contact with teachers and school friends. This helped them to overcome the COVID-19 pandemical period. It was concluded that the students with good Internet access suffered less than those who did not have good Internet access. Also, aspects like teaching methods and principles, didactic and techniques organizational forms can affect students approach to learning process [32; 69; 70].

The results show that all teachers, regardless of the school’s type, tried to establish quality, equity, and justice for their students during the COVID-19 pandemical period and these actions’ success was perceived. What is expected is that these actions to establish quality, equity, and justice to the students can be expanded to the post-COVID-19 pandemical period and that the education continues to fulfill its role of guaranteeing the full individual development, preparing to citizenship exercise, and qualifying for the job market.

-------------------------------------------------------------------------------------------------------------------------------

 The Reviewer 2 – Comment 5

CONCLUSIONS: They are synthetic and reiterate the results presented. They should be redone.

Comments of the authors about Reviewer 1 – Comment 5

The conclusion was revised as you can see below.

  1. Conclusions and limitation and direction for future studies

The COVID-19 pandemic brought many challenges and one of the greatest ones was related to education. Suddenly, students and teachers had to adapt to a new teaching modality and to the social isolation that interfered with their emotions and feelings. It brought, despite new and positive learning, some counterproductive consequences since it was suddenly imposed, without giving time to those involved to properly assimilate the new methods. The percentage of students who improved their scholar and learning performance was low if compared to the percentages of those who maintained or reduced their scholar and learning performance. These results can be seen, in general, both in the results and in the concept and careful observation of the teachers. However, teachers from private and federal schools demonstrated that there was a less marked reduction in student performance in these schools, that is, this negative result showed a higher percentage in municipal and state schools. All the teachers claimed to have sought to stimulate the learning of these students by seeking various other non-complex and playful activities, like Facebook, WhatsApp, emails, explanatory videos, and making adaptations in the planning trying to understand the reality of the students with solidarity and expanding the variety of materials that were offered. Different and innovative technologies can be applied for teachers to motivate students [69; 70] and for the students it allows to progress on their own individual pace [41]. In addition, the teachers sent motivational messages, to offer quality, equality, and justice in teaching during the COVID-19 pandemical period. In terms of equality, federal schools stood out more positively, with state and municipal schools presenting lower results in quality.

Limitations and Directions for Further Research

This study has limitations regarding the population. Due to the impossibility of interviewing the entire population, the authors of this paper developed a random sample of the universe. Another point of this exploratory research is the fact that it was developed in the city of Rio de Janeiro, which is in the state of Rio de Janeiro in the southeastern region of Brazil. Since Brazil is a continental country, the results cannot be generalized to all Brazilian cities and states.

Also, this study is an exploratory one, it did not plan to check the hypotheses that were created. For future research, the expectation is to verify the directions taken by teaching activities during the COVID-19 pandemical period in other Brazilian regions, other countries, and in the Brazilian and foreign universities. Another research agenda is to verify the effects of the post-COVID-19 pandemic in education and to expand the study on the inclusion of sustainable concepts in the Brazilian schools and universities.

-------------------------------------------------------------------------------------------------------------------------------

 The Reviewer 2 – Comment 6

 BIBLIOGRAPHICAL REFERENCES. Very few and it would be necessary to incorporate more references on the subject studied that have been generated in recent years.

 Comments of the authors about Reviewer 2 – Comment 6

Thank you very much. The references were adjusted, and new references were incorporated as can be seen below:

References

  1. Stanton: R., To, Q. G., Khalesi, S., Williams, S. L., Alley, S. J., Thwaite, T. L., Fenning, A. S., & Vandelanotte, C. (2020). “Depression, Anxiety and Stress during COVID-19: Associations with Changes in Physical Activity, Sleep, Tobacco and Alcohol Use in Australian Adults”. Int. J. Environ. Res. Public Health, Vol. 17, 4065.

  1. Dias, A., Scavarda, A., Reis, A., Silveira, H. & Ebecken, N. F. F. (2020). “Managerial Strategies for Long-Term Care Organization Professionals: COVID-19 Pandemic Impacts”. Sustainability, Vol.12, Number 22. https://doi.org/10.3390/su12229682.

  1. Scavarda, A., Dias, A., Reis, A., Silveira, H., & Santos, I. A. (2021). “COVID-19 Pandemic Sustainable Educational Innovation Management Proposal Framework”. Sustainability, Vol. 13, pp. 6391. https://doi.org/10.3390/su13116391.

  1. Dias, A., Scavarda, A., Silveira, H., Scavarda, L. F., Kondamareddy, K. K. (2021). “The Educational Online System: COVID-19 Demands, Trends, Implications, Challenges, Lessons, Insights, Opportunities, Outlooks, and Directions in the Work from Home”. Sustainability, Vol. 13, pp. 12197. https://doi.org/10.3390/su132112197.

  1. Kaden, U. (2020). “COVID-19 School Closure-Related Changes to the Professional Life of a K–12 Teacher”. Educ. Sci., Vol. 10, pp. 165. https://doi:10.3390/educsci10060165.

  1. Gandolfi, A. (2020). “Planning of school teaching during Covid-19”. Physica, Vol. 415, pp. 132753. https://doi.org/10.1016/j.physd.2020.132753.

  1. Kim, L. E., & Asbury, K. (2020). “Like a rug had been pulled from under you: The impact of COVID-19 on teachers in England during the first six weeks of the UK lockdown”. British Journal of Educational Psychology, Vol. 90, pp. 1062–1083. https://doi:10.1111/bjep.12381.

  1. Kim, C. J. H., & Padilla, A. M. (2020). “Technology for Educational Purposes Among Low-Income Latino Children Living in a Mobile Park in Silicon Valley: A Case Study Before and During COVID-19”. Hispanic Journal of Behavioral Sciences, Vol. 42, Number 4, pp. 497–514. https://doi: 10.1177/0739986320959764.

  1. Pulimeno, M., Piscitelli, P., Colazzo, S., Colao, A., & Miani, A. (2020). “Indoor air quality at school and students’ performance: Recommendations of the UNESCO Chair on Health Education and Sustainable Development & the Italian Society of Environmental Medicine (SIMA)”. Health Promotion Perspectives, Vol. 10, Number 3, pp. 169-174. https://doi: 10.34172/hpp.2020.29.

  1. Helmandollar, M. S. (2020). “Meeting Students Where They Are: Implementing Canvas for Successful Student Outreach”. Inquiry: The Journal of the Virginia Community Colleges, Vol. 23, Number 1. https://commons.vccs.edu/inquiry/vol23/iss1/14.

  1. Svalina, V., & Ivić, V. (2020). “Case study of a student with disabilities in a vocational school during the period of online virtual classes due to Covid-19”. World Journal of Education, Vol. 10, Number 4.  https://doi:10.5430/wje.v10n4p115.

  1. Holt, E. A., Heim, A. B., Tessens, E., & Walker, R. (2020). “Thanks for inviting me to the party: Virtual poster sessions as a way to connect in a time of disconnection”. Ecology and Evolution, Vol. 10, pp. 12423–12430. https://doi: 10.1002/ece3.6756.

  1. Aguilera-Hermida, A. P. (2020). “College students’ use and acceptance of emergency online learning due to COVID-19”. International Journal of Educational Research Open, Vol.1, Number 100011. http://doi: 1016/j.ijedro.2020.100011.

  1. Mishra, L., Gupta, T. & Shree, A. (2020). “Online teaching-learning in higher education during lockdown period of COVID-19 pandemic”. International Journal of Educational Research open, Vol. 1, Number 100012. https://doi.org/10.1016/j.ijedro.2020.100012.

  1. Harries, A.J., Lee, C., Jones, L. et al. (2021). “Effects of the COVID-19 pandemic on medical students: a multicenter quantitative study”. BMC Med Educ, Vol. 21, Number 14. https://doi.org/10.1186/s12909-020-02462-1

  1. Ali, U., Herbst, C. M., & Makridis, C. A. (2021). “The impact of COVID-19 on the U.S. childcare market: Evidence  from stay-at-home orders”. Economics of Education Review, Vol. 82, Number 102094.   https://doi.org/10.1016/j.econedurev.2021.102094.

  1. Code, J., Ralph, R., & Forde, K. (2020). “Pandemic designs for the future: perspectives of technology education teachers during COVID-19”. Information and Learning Sciences, Vol. 121, Number 5/6, pp. 419-431. https://doi 10.1108/ILS-04-2020-0112.

  1. Li, B.Z., Cao, N.W., Ren, C.X., Chu, X.J., Zhou, H.Y., & Guo, B. (2020). “Flipped classroom improves nursing students’ theoretical learning in China: A meta-analysis”. PLoS ONE, Vol. 15, Number 8, pp. https://doi.org/10.1371/journal.pone.0237926.

  1. Abuhammad, S. (2020). “Barriers to distance learning during the COVID-19 outbreak: A qualitative review from parents’ perspective”. Heliyon, Vol. 6, pp. e05482. https://doi.org/10.1016/j.heliyon.2020.e05482.

  1. Armstrong-Mensah, E., Ramsey-White, K., Yankey, B., & Self-Brown, S. (2020). “COVID-19 and Distance Learning: Effects on Georgia State University School of Public Health Students”. Front. Public Health, Vol. 8, pp. 576227. http://doi: 10.3389/fpubh.2020.576227.

  1. Abbasi, J. (2020). “Social Isolation the other COVID-19 threat in nursing homes”. JAMA, Vol. 324, Number 7.

  1. Tran, T., Hoang, A. D., Nguyen, T. T., Dinh, V. H., Nguyen, Y. C., & Pham, H. H. (2020). “Dataset of Vietnamese student’s learning habits during COVID-19”. Data in Brief, Vol. 30 pp. 105682. https://doi.org/10.1016/j.dib.2020.105682.

  1. Tran, T., Hoang, A.D., Nguyen, Y. C., Nguyen, L.C., Ta, N. T., Pham, Q. H., Pham, C. X., L, Q.A., Dinh, V. H., & Nguyen, T. T. (2020). “Toward Sustainable Learning during School Suspension: Socioeconomic, Occupational Aspirations, and Learning Behavior of Vietnamese Students during COVID-19”. Sustainability Vol. 12, pp. 4195. http://doi:10.3390/su12104195.

  1. Dias, A.C., & Reis, A.C. (2017). “Estágio Supervisionado em arquivologia: Pontos fortes e fracos e sugestões para de melhoria para o programa”. Ci. Inf., Vol. 46, pp. 84–105. https://doi.org 0.18225/ci.inf..v46i2.4145.

  1. Cutri, R. M., Mena, J., & Whiting, E. F. (2020). “Faculty readiness for online crisis teaching: transitioning to online teaching during the COVID-19 pandemic”. European Journal of Teacher Education, Vol. 43, Number 4, pp. 523–541. https://doi.org/10.1080/02619768.2020.1815702.
  2. Daú, G., Scavarda, A., Scavarda, L. F., & Portugal, V. J. T. (2019). “The Healthcare Sustainable Supply Chain 4.0: The Circular Economy Transition Conceptual Framework with the Corporate Social Responsibility Mirror”. Sustainability, Vol. 11, Number 18, pp. 5130. https://doi.org/10.3390/su11185130.

27.     Karademir, A., Yaman, F., & Saatçioğlu, Ö. (2020). “Challenges of higher education institutions against COVID-19: The case of Turkey”. Journal of Pedagogical Research, Vol. 4, Number 4, pp. 453-474.  http://dx.doi.org/10.33902/JPR.2020063574.

  1. Olena, O., Nguyen, T. K., & Balakrishnan, V. D. (2020). “International students in Australia – during and after COVID-19”. Higher Education Research & Development, Vol. 39, Number 7, pp. 1372-1376, DOI: 10.1080/07294360.2020.1825346.

  1. Almusharraf, N. M., & Khahro, S. H. (2020). “Students’ Satisfaction with Online Learning Experiences During the COVID-19 Pandemic”. International Journal of Emerging Technologies in Learning, Vol. 15, Number 21, 2020. https://doi.org/10.3991/ijet.v15i21.15647.

  1. König, J., Jäger-Biela, D. J., & Glutsch, N. (2020). “Adapting to online teaching during COVID-19 school closure: teacher education and teacher competence effects among early career teachers in Germany”, European Journal of Teacher Education, Vol. 43, Number 4, pp. 608-622, DOI: 10.1080/02619768.2020.1809650.

  1. Bolumole, M. (2020). “Student life in the age of COVID-19”, Higher Education Research & Development, Vol. 39, Number 7, pp. 1357-1361, DOI: 10.1080/07294360.2020.1825345.

  1. Pavlíková, M., Sirotkin, A., Králik, R., Petrikovičová, L., & Martin, J.G. How to Keep University Active during COVID-19 Pandemic: Experience from Slovakia. Sustainability 2021, 13, 10350. https://doi.org/10.3390/su131810350

  1. McGill, L. (2020). “Start-up company: how and why universities should nurture student friendships from day one”, Perspectives: Policy and Practice in Higher Education, Vol. 24, Number 1, pp. 4-7.

  1. Berwick, D. M. (2020). “COVID-19: Beyond Tomorrow: Choices for the “New Normal””, JAMA, Vol. 323 Number 21, pp. 2125-2126.

  1. Moorhouse, B. L. (2020). “Adaptations to a face-to-face initial teacher education course ‘forced’ online due to the COVID-19 pandemic”, Journal of Education for Teaching, Vol. 46, Number 4, pp. 609-611, DOI: 10.1080/02607476.2020.1755205.

  1. Rajhans, V., Memon, U., Patil, V., & Goyal, A. (2020). “Impact of COVID-19 on academic activities and way forward in Indian Optometry”. Journal of Optometry, Vol. 13, pp. 216-226.

  1. Lee, K., Fanguy, M., Lu, X. S., & Bligh, B. (2021). “Student learning during COVID-19: It was not as bad as we feared”, Distance Education, DOI: 10.1080/01587919.2020.1869529.

  1. Fischer, G., Lundin, J., & Lindberg, J. Ola. (2020). “Rethinking and reinventing learning, education and collaboration in the digital age—from creating technologies to transforming cultures”. The International Journal of Information and Learning Technology, Vol. 37, Number 5, pp. 241-252. DOI 10.1108/IJILT-04-2020-0051.

  1. Lacka, E., Wong, T. C., & Haddoud, M. Y. (2021). “Can digital technologies improve students’ efficiency? Exploring the role of Virtual Learning Environment and Social Media use in Higher Education”. Computers & Education, Vol. 163, pp. 104099. https://doi.org/10.1016/j.compedu.2020.104099.

  1. Šolcová, L., & Magdin, M. Interactive textbook - A new tool in off-line and on-line education. Turkish Online Journal of Educational Technology. 2016, 15 (3).

  1. Lewis, Peter, Osborne, Yvonne, Gray, Genevieve, & Lacaze, Anne-Marie. Design and delivery of a distance education programme: Educating Vietnamese nurse academics from Australia. Procedia: Social and Behavioral Sciences, 47, pp. 1462-1468. https://eprints.qut.edu.au/57259/

  1. Hašková, A., Šafranko, C., Pavlíková, M., & Petrikovičová, L. Application of online teaching tools and aids during corona pandemics. Ad Alta: Journal of Interdisciplinary Research, 2020, 10(2), 106-112,  https://doi.org/10.33543/1002.

  1. Kobylarek, A., Błaszczyński, K., Ślósarz, L., Madej, M., Carmo, A., Hlad, Ľ., Králik, R., Akimjak, A., Judák, V., Maturkanič, P., Biryukova, Y., Tokárová, B., Martin, J.G., & Petrikovičová, L. The Quality of Life among University of the Third Age Students in Poland, Ukraine and Belarus. Sustainability 2022, 14, 2049. https://doi.org/10.3390/su14042049.

44. Tkac, L., & Schauer, F. Dissemination of science among students by remote experimentation. Cyprus International conference on educational research (CY-ICER-2012). Procedia - Social and Behavioral Sciences. 2012, 47, pp.1335-1340. DOI:10.1016/j.sbspro.2012.06.822.

45.       Diniz, V. L., & Silva, R. A. da. (2020). “Formação de professores no Período Pandêmico: (Im)Possibilidades de Ações e Acolhimento no Curso de Geografia da Uft/Araguaína”. Revista Docência do Ensino Superior, Vol. 10, pp. 1-18. DOI: https://doi.org/10.35699/2237-5864.2020.24711.

  1. Moser, K. M., Wei, T., & Brenner, D. (2021). “Remote teaching during COVID-19: Implications from a national survey of language educators”. System, Vol. 97, pp. 102431 https://doi.org/10.1016/j.system.2020.102431.

  1. Sallaberry, J. D. et al. (2020). “Desafios docentes em tempos de isolamento social: estudo com professores do curso de Ciências Contábeis”. Revista Docência do Ensino Superior, Vol. 10, Number e024774, pp. 1-22, 2020. https://doi.org/10.35699/2237-5864.2020.24774.

  1. Radu, Maria-Crina, Schnakovszky, C., Herghelegiu, E., Ciubotariu, Vlad-Andrei, & Cristea, I. The Impact of the COVID-19 Pandemic on the Quality of Educational Process: A Student Survey. Int. J. Environ. Res. Public Health 2020, 17, 7770; doi:10.3390/ijerph17217770.

  1. Mok, Ka Ho, Xiong, Weiyan, Ke, Guoguo, Cheung, & Joyce Oi Wun. (2021). “Impact of COVID-19 pandemic on international higher education and student mobility: Student perspectives from mainland China and Hong Kong”. International Journal of Educational Research, Vol. 105, Number 101718.  https://doi.org/10.1016/j.ijer.2020.101718.

  1. Andrew, A., Cattan, S., Dias, M. C., Farquharson, C., Kraftman, L., Krutikova, S., Phimister, A., & Sevilla, A. (2020). “Inequalities in Children’s Experiences of Home Learning during the COVID-19 Lockdown in England”. Fiscal Studies, Vol. 41, Number 3, pp. 653–683.

  1. Andrew, A., Cattan, S., Dias, M. C., Farquharson, C., Kraftman, L., Krutikova, S., Phimister, A.,  & Sevilla, A. Inequalities in Children’s Experiences of Home Learning during the COVID-19 Lockdown in England. FISCAL STUDIES, vol. 41, no. 3, pp. 653–683 (2020) 0143-5671.

  1. Jiménez Hernández, A.S., Cáceres-Muñoz, J., & Martín-Sánchez, M. Social Justice, Participation and School during the COVID-19—The International Project Gira por la Infancia. Sustainability 2021, 13, 2704. https://doi.org/10.3390/su13052704.

  1. Scheffers, F., Moonen, X., & Vugt, E. v. Assessing the quality of support and discovering sources of resilience during COVID-19 measures in people with intellectual disabilities by professional carers. Research in Developmental Disabilities 111 (2021) 103889. https://doi.org/10.1016/j.ridd.2021.103889.

  1. Gozzelino, G., & Matera, F. Pedagogical lines and critical consciousness for quality education at the time of the Covid-19 pandemic.  Form@re - Open Journal per la formazione in rete.  21, n. 3, pp. 191-199 2021. DOI: https://doi.org/10.36253/form-10178.

  1. Utunen, ,  Van Kerkhove, Maria D,  Tokar, A., O'Connell, G., Gamhewage, G., & Socé, I. One Year of Pandemic Learning Response: Benefits of Massive Online Delivery of the World Health Organization’s Technical Guidance. JMIR Public Health Surveill 2021 | vol. 7 | iss. 4 | e28945 | p. 3, https://publichealth.jmir.org/2021/4/e28945

  1. Miyah, Y., Benjelloun, M., Lairini, S., & Lahrichi, A.  COVID-19 Impact on Public Health, Environment, Human Psychology, Global Socioeconomy, and Education. Hindawi. Volume 2022, https://doi.org/10.1155/2022/5578284.

  1. Deng C, Yang S, Liu Q, Feng S, & Chen C (2021) Sustainable development and health assessment model of higher education in India: A mathematical modeling approach. PLoS ONE 16(12): e0261776. https://doi.org/10.1371/journal.pone.0261776

  1. Wipfli, H., & Withers, M. Engaging youth in global health and social justice: a decade of experience teaching a high school summer course, Global Health Action, 15:1, 1987045, 2022.  DOI: 10.1080/16549716.2021.1987045.

  1. Chan, I. L., Mowson, R., Alonso, J. P., Roberti, J., Contreras, M., & Velandia-González, M. Promoting immunization equity in Latin America and the Caribbean:Case studies, lessons learned, and their implication for COVID-19 vaccine Equity. Vaccine 40 (2022) 1977–1986. https://doi.org/10.1016/j.vaccine.2022.02.051.

  1. Baral, S, Chandler, R., Prieto, R. G., Gupta, S., Mishra, S., & Kulldorff, M. Leveraging epidemiological principles to evaluate Sweden’s COVID-19 Response. Annals of Epidemiology 54 (2021) 21e26.  https://doi.org/10.1016/j.annepidem.2020.11.005

  1.  Dewart, G., Corcoran, L., Thirsk, L., & Petrovic, K. “Nursing education in a pandemic: Academic challenges in response to COVID-19”. Nurse Education Today, Vol. 92, Number 104471. https://doi.org/10.1016/j.nedt.2020.104471.

  1. Torda, A. (2020). “How COVID-19 has pushed us into a medical education Revolution”. Internal Medicine Journal. Vol. 50, pp. 1150–1153. https://doi:10.1111/imj.14882.

  1. Tracey J. N. & Tolan, M. (2020). “Online Accounting Courses: Transition and Emerging Issues”. The CPA Journal. Vol. 90, Number 5, pp. 11.

  1. inf.br. (2018). Censo (2018). available at: http://www.escolas.inf.br/rj/rio-de-janeiro. (Accessed 14 August 2020).

  1. Triola, M. F. (2008). Introdução à Estatística. 10.a Edição. LTC- Rio de Janeiro.

  1. Favero, L. P., Belfiore, P., Silva, F. L., & Chan, B. L. (2009). Análise de dados: modelagem multivariada para tomada de decisões. Rio de Janeiro: Elsevier.

  1. Medronho, R. A., Bloch, K.V., Luiz, R. R., & Werneck, G. L. (2009).  São Paulo. Editora Atheneu.

  1. Petrovič, F., Murgaš, F., & Králik, R. (2021). Happiness in Czechia during the COVID-19 Pandemic. Sustainability, 13(19), 10826. https://doi.org/10.3390/su131910826.

  1. Petrikovičová, L., Dysková S., Pavlíková M., Vasbieva D.G., & Kalugina O. A. Teaching geographical methods and forms in the United States, Iceland and Slovakia. Ad Alta: Journal of Interdisciplinary Research, 2021, 11(1), 398-401, https://doi.org/10.33543/1101.

  1. Petrikovičová, L., Ďurinková, A., Králik, R., & Kurilenko, V. Methodology of Working with a Textbook Versus Field Activities of Teaching Geography during the Corona Crisis.  European Journal of Contemporary Education, 2021, 10(2), 428-437,  https://doi.org/10.13187/ejced.2021.2.428.

Reviewer 3 Report

Review Report

Article title: Equity, Justice, and Quality During the COVID-19 Pandemical Period: Considerations on Learning and Scholar Performance in the Brazilian Schools

The author(s) aim to analyze the direction of teaching activities in public and private schools in the city of Rio de Janeiro (Brazil) and their consequences on learning and student performance concerning elementary and middle schools during the COVID-19 pandemic.

This study verified by email questionnaire if there was equality, justice, and quality in teaching methods during the COVID-19 pandemic. The study used the SPSS methodology (Statistical Package for the Social Sciences), version 22.0. The author/authors has/have made a significant effort in guiding the reader through the overall empirical study. However, major issues related to clarity and contribution of this study to the scholarly literature remain to be addressed. The issues are detailed below.

Specific comments:

  1. The text of the paper does not provide the exact wording of the research hypotheses.
  2. Did teachers evaluate the scholar and learning performance of their students (Table 1 and Table 2) only according to their own feelings, or did they also rely on didactic tests results as well in their answers? What is the informative value of the research results based on the feelings of the respondents only?
  3. I think it is a pity that the authors of the article did not use the questionnaire to find out what teaching methods the teachers used during the COVID-19 pandemic period.
  4. There are no pictures or graphs from the research in the text of the article.
  5. The authors do not state whether the science teachers used real remote experiments, interactive simulations, or applets on the Internet in their teaching. Real remote experiments, interactive simulations, and applets on the Internet can significantly contribute to the improvement of teaching results in science during distance learning.

For example: Tkac, L. Schauer, F. Dissemination of science among students by remote experimentation. Cyprus International conference on educational research (CY-ICER-2012). Procedia - Social and Behavioral Sciences. 2012, 47, pp.1335-1340. DOI:10.1016/j.sbspro.2012.06.822.

  1. The conclusions are brief and general. The section needs to be better unfolded, highlighting how the results of the overall study benefit the teaching practice.
  2. References are not numbered in the list.

Author Response

REVIEWER 3

Dear Reviewer,

We thank you very much for all your comments. Please, see below our answers to your questions and the suggested changes already done.

Kind regards,

The Authors

-------------------------------------------------------------------------------------------------------------------------------

Comments and Suggestions for Authors

The Reviewer 3 – Comment 1

Review Report

Article title: Equity, Justice, and Quality During the COVID-19 Pandemical Period: Considerations on Learning and Scholar Performance in the Brazilian Schools

The author(s) aim to analyze the direction of teaching activities in public and private schools in the city of Rio de Janeiro (Brazil) and their consequences on learning and student performance concerning elementary and middle schools during the COVID-19 pandemic.

This study verified by email questionnaire if there was equality, justice, and quality in teaching methods during the COVID-19 pandemic. The study used the SPSS methodology (Statistical Package for the Social Sciences), version 22.0. The author/authors has/have made a significant effort in guiding the reader through the overall empirical study. However, major issues related to clarity and contribution of this study to the scholarly literature remain to be addressed. The issues are detailed below.

Comments of the authors about Reviewer 3 – Comment 1

Thank you very much. We will make all the effort to meet all suggested recommendations.

-------------------------------------------------------------------------------------------------------------------------------

The Reviewer 3 – Comment 2

  1. The text of the paper does not provide the exact wording of the research hypotheses.

Comments of the authors about Reviewer 3 – Comment 2

Thank you very much. The sentence was included in the paper.

Also, this study is an exploratory one, it did not plan to check the hypotheses that were created. For future research, the expectation is to verify the directions taken by teaching activities during the COVID-19 pandemical period in other Brazilian regions, other countries, and in the Brazilian and foreign universities. Another research agenda is to verify the effects of the post-COVID-19 pandemic in education and to expand the study on the inclusion of sustainable concepts in the Brazilian schools and universities.

-------------------------------------------------------------------------------------------------------------------------------

The Reviewer 3 – Comment 3

  1. The authors do not state whether the science teachers used real remote experiments, interactive simulations, or applets on the Internet in their teaching. Real remote experiments, interactive simulations, and applets on the Internet can significantly contribute to the improvement of teaching results in science during distance learning.

For example: Tkac, L. Schauer, F. Dissemination of science among students by remote experimentation. Cyprus International conference on educational research (CY-ICER-2012). Procedia - Social and Behavioral Sciences. 2012, 47, pp.1335-1340. DOI:10.1016/j.sbspro.2012.06.822.

Comments of the authors about Reviewer 3 – Comment 3

Thank you very much for your suggestion. The references below were added.

Šolcová, L.; Magdin, M. Interactive textbook - A new tool in off-line and on-line education. Turkish Online Journal of Educational Technology. 2016, 15,(3)

Hašková, A.; Šafranko, C.; Pavlíková, M.; Petrikovičová, L. Application of online teaching tools and aids during corona pandemics. Ad Alta: Journal of Interdisciplinary Research, 2020, 10(2), 106-112,  https://doi.org/10.33543/1002.

Petrikovičová, L.; Dysková S.; Pavlíková M.; Vasbieva DG.; Kalugina OA. Teaching geographical methods and forms in the United States, Iceland and Slovakia. Ad Alta: Journal of Interdisciplinary Research, 2021, 11(1), 398-401, https://doi.org/10.33543/1101.

Petrikovičová, L.; Ďurinková, A.; Králik, R.; Kurilenko, V. Methodology of Working with a Textbook Versus Field Activities of Teaching Geography during the Corona Crisis.  European Journal of Contemporary Education, 2021, 10(2), 428-437,  https://doi.org/10.13187/ejced.2021.2.428.

Petrovič, F., Murgaš, F., & Králik, R. (2021). Happiness in Czechia during the COVID-19 Pandemic. Sustainability, 13(19), 10826. https://doi.org/10.3390/su131910826

Kobylarek, A.; Błaszczyński, K.; Ślósarz, L.; Madej, M.; Carmo, A.; Hlad, Ľ.; Králik, R.; Akimjak, A.; Judák, V.; Maturkanič, P.; Biryukova, Y.; Tokárová, B.; Martin, J.G.; Petrikovičová, L. The Quality of Life among University of the Third Age Students in Poland, Ukraine and Belarus. Sustainability 2022, 14, 2049. https://doi.org/10.3390/su14042049

Tkac, L. Schauer, F. Dissemination of science among students by remote experimentation. Cyprus International conference on educational research (CY-ICER-2012). Procedia - Social and Behavioral Sciences. 2012, 47, pp.1335-1340. DOI:10.1016/j.sbspro.2012.06.822

-------------------------------------------------------------------------------------------------------------------------------

 The Reviewer 3 – Comment 4

  1. The conclusions are brief and general. The section needs to be better unfolded, highlighting how the results of the overall study benefit the teaching practice.

Comments of the authors about Reviewer 3 – Comment 4

The conclusion was revised as you can see below.

  1. Conclusions and limitation and direction for future studies

The COVID-19 pandemic brought many challenges and one of the greatest ones was related to education. Suddenly, students and teachers had to adapt to a new teaching modality and to the social isolation that interfered with their emotions and feelings. It brought, despite new and positive learning, some counterproductive consequences since it was suddenly imposed, without giving time to those involved to properly assimilate the new methods. The percentage of students who improved their scholar and learning performance was low if compared to the percentages of those who maintained or reduced their scholar and learning performance. These results can be seen, in general, both in the results and in the concept and careful observation of the teachers. However, teachers from private and federal schools demonstrated that there was a less marked reduction in student performance in these schools, that is, this negative result showed a higher percentage in municipal and state schools. All the teachers claimed to have sought to stimulate the learning of these students by seeking various other non-complex and playful activities, like Facebook, WhatsApp, emails, explanatory videos, and making adaptations in the planning trying to understand the reality of the students with solidarity and expanding the variety of materials that were offered. Different and innovative technologies can be applied for teachers to motivate students [69; 70] and for the students it allows to progress on their own individual pace [41]. In addition, the teachers sent motivational messages, to offer quality, equality, and justice in teaching during the COVID-19 pandemical period. In terms of equality, federal schools stood out more positively, with state and municipal schools presenting lower results in quality.

Limitations and Directions for Further Research

This study has limitations regarding the population. Due to the impossibility of interviewing the entire population, the authors of this paper developed a random sample of the universe. Another point of this exploratory research is the fact that it was developed in the city of Rio de Janeiro, which is in the state of Rio de Janeiro in the southeastern region of Brazil. Since Brazil is a continental country, the results cannot be generalized to all Brazilian cities and states.

Also, this study is an exploratory one, it did not plan to check the hypotheses that were created. For future research, the expectation is to verify the directions taken by teaching activities during the COVID-19 pandemical period in other Brazilian regions, other countries, and in the Brazilian and foreign universities. Another research agenda is to verify the effects of the post-COVID-19 pandemic in education and to expand the study on the inclusion of sustainable concepts in the Brazilian schools and universities.

-------------------------------------------------------------------------------------------------------------------------------

The Reviewer 3 – Comment 5

  1. References are not numbered in the list

Comments of the authors about Reviewer 3 – Comment 5

Thank you very much. The references were adjusted as can be seen below.

.

References

  1. Stanton: R., To, Q. G., Khalesi, S., Williams, S. L., Alley, S. J., Thwaite, T. L., Fenning, A. S., & Vandelanotte, C. (2020). “Depression, Anxiety and Stress during COVID-19: Associations with Changes in Physical Activity, Sleep, Tobacco and Alcohol Use in Australian Adults”. Int. J. Environ. Res. Public Health, Vol. 17, 4065.

  1. Dias, A., Scavarda, A., Reis, A., Silveira, H. & Ebecken, N. F. F. (2020). “Managerial Strategies for Long-Term Care Organization Professionals: COVID-19 Pandemic Impacts”. Sustainability, Vol.12, Number 22. https://doi.org/10.3390/su12229682.

  1. Scavarda, A., Dias, A., Reis, A., Silveira, H., & Santos, I. A. (2021). “COVID-19 Pandemic Sustainable Educational Innovation Management Proposal Framework”. Sustainability, Vol. 13, pp. 6391. https://doi.org/10.3390/su13116391.

  1. Dias, A., Scavarda, A., Silveira, H., Scavarda, L. F., Kondamareddy, K. K. (2021). “The Educational Online System: COVID-19 Demands, Trends, Implications, Challenges, Lessons, Insights, Opportunities, Outlooks, and Directions in the Work from Home”. Sustainability, Vol. 13, pp. 12197. https://doi.org/10.3390/su132112197.

  1. Kaden, U. (2020). “COVID-19 School Closure-Related Changes to the Professional Life of a K–12 Teacher”. Educ. Sci., Vol. 10, pp. 165. https://doi:10.3390/educsci10060165.

  1. Gandolfi, A. (2020). “Planning of school teaching during Covid-19”. Physica, Vol. 415, pp. 132753. https://doi.org/10.1016/j.physd.2020.132753.

  1. Kim, L. E., & Asbury, K. (2020). “Like a rug had been pulled from under you: The impact of COVID-19 on teachers in England during the first six weeks of the UK lockdown”. British Journal of Educational Psychology, Vol. 90, pp. 1062–1083. https://doi:10.1111/bjep.12381.

  1. Kim, C. J. H., & Padilla, A. M. (2020). “Technology for Educational Purposes Among Low-Income Latino Children Living in a Mobile Park in Silicon Valley: A Case Study Before and During COVID-19”. Hispanic Journal of Behavioral Sciences, Vol. 42, Number 4, pp. 497–514. https://doi: 10.1177/0739986320959764.

  1. Pulimeno, M., Piscitelli, P., Colazzo, S., Colao, A., & Miani, A. (2020). “Indoor air quality at school and students’ performance: Recommendations of the UNESCO Chair on Health Education and Sustainable Development & the Italian Society of Environmental Medicine (SIMA)”. Health Promotion Perspectives, Vol. 10, Number 3, pp. 169-174. https://doi: 10.34172/hpp.2020.29.

  1. Helmandollar, M. S. (2020). “Meeting Students Where They Are: Implementing Canvas for Successful Student Outreach”. Inquiry: The Journal of the Virginia Community Colleges, Vol. 23, Number 1. https://commons.vccs.edu/inquiry/vol23/iss1/14.

  1. Svalina, V., & Ivić, V. (2020). “Case study of a student with disabilities in a vocational school during the period of online virtual classes due to Covid-19”. World Journal of Education, Vol. 10, Number 4.  https://doi:10.5430/wje.v10n4p115.

  1. Holt, E. A., Heim, A. B., Tessens, E., & Walker, R. (2020). “Thanks for inviting me to the party: Virtual poster sessions as a way to connect in a time of disconnection”. Ecology and Evolution, Vol. 10, pp. 12423–12430. https://doi: 10.1002/ece3.6756.

  1. Aguilera-Hermida, A. P. (2020). “College students’ use and acceptance of emergency online learning due to COVID-19”. International Journal of Educational Research Open, Vol.1, Number 100011. http://doi: 1016/j.ijedro.2020.100011.

  1. Mishra, L., Gupta, T. & Shree, A. (2020). “Online teaching-learning in higher education during lockdown period of COVID-19 pandemic”. International Journal of Educational Research open, Vol. 1, Number 100012. https://doi.org/10.1016/j.ijedro.2020.100012.

  1. Harries, A.J., Lee, C., Jones, L. et al. (2021). “Effects of the COVID-19 pandemic on medical students: a multicenter quantitative study”. BMC Med Educ, Vol. 21, Number 14. https://doi.org/10.1186/s12909-020-02462-1

  1. Ali, U., Herbst, C. M., & Makridis, C. A. (2021). “The impact of COVID-19 on the U.S. childcare market: Evidence  from stay-at-home orders”. Economics of Education Review, Vol. 82, Number 102094.   https://doi.org/10.1016/j.econedurev.2021.102094.

  1. Code, J., Ralph, R., & Forde, K. (2020). “Pandemic designs for the future: perspectives of technology education teachers during COVID-19”. Information and Learning Sciences, Vol. 121, Number 5/6, pp. 419-431. https://doi 10.1108/ILS-04-2020-0112.

  1. Li, B.Z., Cao, N.W., Ren, C.X., Chu, X.J., Zhou, H.Y., & Guo, B. (2020). “Flipped classroom improves nursing students’ theoretical learning in China: A meta-analysis”. PLoS ONE, Vol. 15, Number 8, pp. https://doi.org/10.1371/journal.pone.0237926.

  1. Abuhammad, S. (2020). “Barriers to distance learning during the COVID-19 outbreak: A qualitative review from parents’ perspective”. Heliyon, Vol. 6, pp. e05482. https://doi.org/10.1016/j.heliyon.2020.e05482.

  1. Armstrong-Mensah, E., Ramsey-White, K., Yankey, B., & Self-Brown, S. (2020). “COVID-19 and Distance Learning: Effects on Georgia State University School of Public Health Students”. Front. Public Health, Vol. 8, pp. 576227. http://doi: 10.3389/fpubh.2020.576227.

  1. Abbasi, J. (2020). “Social Isolation the other COVID-19 threat in nursing homes”. JAMA, Vol. 324, Number 7.

  1. Tran, T., Hoang, A. D., Nguyen, T. T., Dinh, V. H., Nguyen, Y. C., & Pham, H. H. (2020). “Dataset of Vietnamese student’s learning habits during COVID-19”. Data in Brief, Vol. 30 pp. 105682. https://doi.org/10.1016/j.dib.2020.105682.

  1. Tran, T., Hoang, A.D., Nguyen, Y. C., Nguyen, L.C., Ta, N. T., Pham, Q. H., Pham, C. X., L, Q.A., Dinh, V. H., & Nguyen, T. T. (2020). “Toward Sustainable Learning during School Suspension: Socioeconomic, Occupational Aspirations, and Learning Behavior of Vietnamese Students during COVID-19”. Sustainability Vol. 12, pp. 4195. http://doi:10.3390/su12104195.

  1. Dias, A.C., & Reis, A.C. (2017). “Estágio Supervisionado em arquivologia: Pontos fortes e fracos e sugestões para de melhoria para o programa”. Inf., Vol. 46, pp. 84–105. https://doi.org 0.18225/ci.inf..v46i2.4145.

  1. Cutri, R. M., Mena, J., & Whiting, E. F. (2020). “Faculty readiness for online crisis teaching: transitioning to online teaching during the COVID-19 pandemic”. European Journal of Teacher Education, Vol. 43, Number 4, pp. 523–541. https://doi.org/10.1080/02619768.2020.1815702.
  2. Daú, G., Scavarda, A., Scavarda, L. F., & Portugal, V. J. T. (2019). “The Healthcare Sustainable Supply Chain 4.0: The Circular Economy Transition Conceptual Framework with the Corporate Social Responsibility Mirror”. Sustainability, Vol. 11, Number 18, pp. 5130. https://doi.org/10.3390/su11185130.

27.     Karademir, A., Yaman, F., & Saatçioğlu, Ö. (2020). “Challenges of higher education institutions against COVID-19: The case of Turkey”. Journal of Pedagogical Research, Vol. 4, Number 4, pp. 453-474.  http://dx.doi.org/10.33902/JPR.2020063574.

  1. Olena, O., Nguyen, T. K., & Balakrishnan, V. D. (2020). “International students in Australia – during and after COVID-19”. Higher Education Research & Development, Vol. 39, Number 7, pp. 1372-1376, DOI: 10.1080/07294360.2020.1825346.

  1. Almusharraf, N. M., & Khahro, S. H. (2020). “Students’ Satisfaction with Online Learning Experiences During the COVID-19 Pandemic”. International Journal of Emerging Technologies in Learning, Vol. 15, Number 21, 2020. https://doi.org/10.3991/ijet.v15i21.15647.

  1. König, J., Jäger-Biela, D. J., & Glutsch, N. (2020). “Adapting to online teaching during COVID-19 school closure: teacher education and teacher competence effects among early career teachers in Germany”, European Journal of Teacher Education, Vol. 43, Number 4, pp. 608-622, DOI: 10.1080/02619768.2020.1809650.

  1. Bolumole, M. (2020). “Student life in the age of COVID-19”, Higher Education Research & Development, Vol. 39, Number 7, pp. 1357-1361, DOI: 10.1080/07294360.2020.1825345.

  1. Pavlíková, M., Sirotkin, A., Králik, R., Petrikovičová, L., & Martin, J.G. How to Keep University Active during COVID-19 Pandemic: Experience from Slovakia. Sustainability 2021, 13, 10350. https://doi.org/10.3390/su131810350

  1. McGill, L. (2020). “Start-up company: how and why universities should nurture student friendships from day one”, Perspectives: Policy and Practice in Higher Education, Vol. 24, Number 1, pp. 4-7.

  1. Berwick, D. M. (2020). “COVID-19: Beyond Tomorrow: Choices for the “New Normal””, JAMA, Vol. 323 Number 21, pp. 2125-2126.

  1. Moorhouse, B. L. (2020). “Adaptations to a face-to-face initial teacher education course ‘forced’ online due to the COVID-19 pandemic”, Journal of Education for Teaching, Vol. 46, Number 4, pp. 609-611, DOI: 10.1080/02607476.2020.1755205.

  1. Rajhans, V., Memon, U., Patil, V., & Goyal, A. (2020). “Impact of COVID-19 on academic activities and way forward in Indian Optometry”. Journal of Optometry, Vol. 13, pp. 216-226.

  1. Lee, K., Fanguy, M., Lu, X. S., & Bligh, B. (2021). “Student learning during COVID-19: It was not as bad as we feared”, Distance Education, DOI: 10.1080/01587919.2020.1869529.

  1. Fischer, G., Lundin, J., & Lindberg, J. Ola. (2020). “Rethinking and reinventing learning, education and collaboration in the digital age—from creating technologies to transforming cultures”. The International Journal of Information and Learning Technology, Vol. 37, Number 5, pp. 241-252. DOI 10.1108/IJILT-04-2020-0051.

  1. Lacka, E., Wong, T. C., & Haddoud, M. Y. (2021). “Can digital technologies improve students’ efficiency? Exploring the role of Virtual Learning Environment and Social Media use in Higher Education”. Computers & Education, Vol. 163, pp. 104099. https://doi.org/10.1016/j.compedu.2020.104099.

  1. Šolcová, L., & Magdin, M. Interactive textbook - A new tool in off-line and on-line education. Turkish Online Journal of Educational Technology. 2016, 15 (3).

  1. Lewis, Peter, Osborne, Yvonne, Gray, Genevieve, & Lacaze, Anne-Marie. Design and delivery of a distance education programme: Educating Vietnamese nurse academics from Australia. Procedia: Social and Behavioral Sciences, 47, pp. 1462-1468. https://eprints.qut.edu.au/57259/

  1. Hašková, A., Šafranko, C., Pavlíková, M., & Petrikovičová, L. Application of online teaching tools and aids during corona pandemics. Ad Alta: Journal of Interdisciplinary Research, 2020, 10(2), 106-112,  https://doi.org/10.33543/1002.

  1. Kobylarek, A., Błaszczyński, K., Ślósarz, L., Madej, M., Carmo, A., Hlad, Ľ., Králik, R., Akimjak, A., Judák, V., Maturkanič, P., Biryukova, Y., Tokárová, B., Martin, J.G., & Petrikovičová, L. The Quality of Life among University of the Third Age Students in Poland, Ukraine and Belarus. Sustainability 2022, 14, 2049. https://doi.org/10.3390/su14042049.

44.     Tkac, L., & Schauer, F. Dissemination of science among students by remote experimentation. Cyprus International conference on educational research (CY-ICER-2012). Procedia - Social and Behavioral Sciences. 2012, 47, pp.1335-1340. DOI:10.1016/j.sbspro.2012.06.822.

45.       Diniz, V. L., & Silva, R. A. da. (2020). “Formação de professores no Período Pandêmico: (Im)Possibilidades de Ações e Acolhimento no Curso de Geografia da Uft/Araguaína”. Revista Docência do Ensino Superior, Vol. 10, pp. 1-18. DOI: https://doi.org/10.35699/2237-5864.2020.24711.

  1. Moser, K. M., Wei, T., & Brenner, D. (2021). “Remote teaching during COVID-19: Implications from a national survey of language educators”. System, Vol. 97, pp. 102431 https://doi.org/10.1016/j.system.2020.102431.

  1. Sallaberry, J. D. et al. (2020). “Desafios docentes em tempos de isolamento social: estudo com professores do curso de Ciências Contábeis”. Revista Docência do Ensino Superior, Vol. 10, Number e024774, pp. 1-22, 2020. https://doi.org/10.35699/2237-5864.2020.24774.

  1. Radu, Maria-Crina, Schnakovszky, C., Herghelegiu, E., Ciubotariu, Vlad-Andrei, & Cristea, I. The Impact of the COVID-19 Pandemic on the Quality of Educational Process: A Student Survey. Int. J. Environ. Res. Public Health 2020, 17, 7770; doi:10.3390/ijerph17217770.

  1. Mok, Ka Ho, Xiong, Weiyan, Ke, Guoguo, Cheung, & Joyce Oi Wun. (2021). “Impact of COVID-19 pandemic on international higher education and student mobility: Student perspectives from mainland China and Hong Kong”. International Journal of Educational Research, Vol. 105, Number 101718.  https://doi.org/10.1016/j.ijer.2020.101718.

  1. Andrew, A., Cattan, S., Dias, M. C., Farquharson, C., Kraftman, L., Krutikova, S., Phimister, A., & Sevilla, A. (2020). “Inequalities in Children’s Experiences of Home Learning during the COVID-19 Lockdown in England”. Fiscal Studies, Vol. 41, Number 3, pp. 653–683.

  1. Andrew, A., Cattan, S., Dias, M. C., Farquharson, C., Kraftman, L., Krutikova, S., Phimister, A.,  & Sevilla, A. Inequalities in Children’s Experiences of Home Learning during the COVID-19 Lockdown in England. FISCAL STUDIES, vol. 41, no. 3, pp. 653–683 (2020) 0143-5671.

  1. Jiménez Hernández, A.S., Cáceres-Muñoz, J., & Martín-Sánchez, M. Social Justice, Participation and School during the COVID-19—The International Project Gira por la Infancia. Sustainability 2021, 13, 2704. https://doi.org/10.3390/su13052704.

  1. Scheffers, F., Moonen, X., & Vugt, E. v. Assessing the quality of support and discovering sources of resilience during COVID-19 measures in people with intellectual disabilities by professional carers. Research in Developmental Disabilities 111 (2021) 103889. https://doi.org/10.1016/j.ridd.2021.103889.

  1. Gozzelino, G., & Matera, F. Pedagogical lines and critical consciousness for quality education at the time of the Covid-19 pandemic.  Form@re - Open Journal per la formazione in rete.  21, n. 3, pp. 191-199 2021. DOI: https://doi.org/10.36253/form-10178.

  1. Utunen, ,  Van Kerkhove, Maria D,  Tokar, A., O'Connell, G., Gamhewage, G., & Socé, I. One Year of Pandemic Learning Response: Benefits of Massive Online Delivery of the World Health Organization’s Technical Guidance. JMIR Public Health Surveill 2021 | vol. 7 | iss. 4 | e28945 | p. 3, https://publichealth.jmir.org/2021/4/e28945

  1. Miyah, Y., Benjelloun, M., Lairini, S., & Lahrichi, A.  COVID-19 Impact on Public Health, Environment, Human Psychology, Global Socioeconomy, and Education. Hindawi. Volume 2022, https://doi.org/10.1155/2022/5578284.

  1. Deng C, Yang S, Liu Q, Feng S, & Chen C (2021) Sustainable development and health assessment model of higher education in India: A mathematical modeling approach. PLoS ONE 16(12): e0261776. https://doi.org/10.1371/journal.pone.0261776

  1. Wipfli, H., & Withers, M. Engaging youth in global health and social justice: a decade of experience teaching a high school summer course, Global Health Action, 15:1, 1987045, 2022.  DOI: 10.1080/16549716.2021.1987045.

  1. Chan, I. L., Mowson, R., Alonso, J. P., Roberti, J., Contreras, M., & Velandia-González, M. Promoting immunization equity in Latin America and the Caribbean:Case studies, lessons learned, and their implication for COVID-19 vaccine Equity. Vaccine 40 (2022) 1977–1986. https://doi.org/10.1016/j.vaccine.2022.02.051.

  1. Baral, S, Chandler, R., Prieto, R. G., Gupta, S., Mishra, S., & Kulldorff, M. Leveraging epidemiological principles to evaluate Sweden’s COVID-19 Response. Annals of Epidemiology 54 (2021) 21e26.  https://doi.org/10.1016/j.annepidem.2020.11.005

  1.  Dewart, G., Corcoran, L., Thirsk, L., & Petrovic, K. “Nursing education in a pandemic: Academic challenges in response to COVID-19”. Nurse Education Today, Vol. 92, Number 104471. https://doi.org/10.1016/j.nedt.2020.104471.

  1. Torda, A. (2020). “How COVID-19 has pushed us into a medical education Revolution”. Internal Medicine Journal. Vol. 50, pp. 1150–1153. https://doi:10.1111/imj.14882.

  1. Tracey J. N. & Tolan, M. (2020). “Online Accounting Courses: Transition and Emerging Issues”. The CPA Journal. Vol. 90, Number 5, pp. 11.

  1. inf.br. (2018). Censo (2018). available at: http://www.escolas.inf.br/rj/rio-de-janeiro. (Accessed 14 August 2020).

  1. Triola, M. F. (2008). Introdução à Estatística. 10.a Edição. LTC- Rio de Janeiro.

  1. Favero, L. P., Belfiore, P., Silva, F. L., & Chan, B. L. (2009). Análise de dados: modelagem multivariada para tomada de decisões. Rio de Janeiro: Elsevier.

  1. Medronho, R. A., Bloch, K.V., Luiz, R. R., & Werneck, G. L. (2009).  São Paulo. Editora Atheneu.

  1. Petrovič, F., Murgaš, F., & Králik, R. (2021). Happiness in Czechia during the COVID-19 Pandemic. Sustainability, 13(19), 10826. https://doi.org/10.3390/su131910826.

  1. Petrikovičová, L., Dysková S., Pavlíková M., Vasbieva D., & Kalugina O. A. Teaching geographical methods and forms in the United States, Iceland and Slovakia. Ad Alta: Journal of Interdisciplinary Research, 2021, 11(1), 398-401, https://doi.org/10.33543/1101.

  1. Petrikovičová, L., Ďurinková, A., Králik, R., & Kurilenko, V. Methodology of Working with a Textbook Versus Field Activities of Teaching Geography during the Corona Crisis.  European Journal of Contemporary Education, 2021, 10(2), 428-437,  https://doi.org/10.13187/ejced.2021.2.428.

Round 2

Reviewer 2 Report

All comments have been responded to, however, my comments have not been addressed in:
METHODS. The sample is large, teachers are surveyed online. There is no explanation of how the sampling was carried out and whether the informants are representative according to the distribution of school typologies present in the population.
The research questions do not appear and the objective is very generic.
The research methodology is not clearly specified, although the statistical techniques to be used are exhaustively expressed. The characteristics of the questionnaire are not explained, whether it was based on closed questions, had a Likert scale, etc.

Author Response

Comments of the authors about Reviewer 2 – Comment 1

Thank you very much. The methodology was adjusted as can be seen below:

This study had as a population the total number of K-12 teachers in the city of Rio de Janeiro. It was based on the 2018 Schools Census (escolas.inf.br, 2018) [64] that registered 66,999 teachers in the city distributed in 2010 private schools, 1,439 municipal schools, 457 state schools, and 28 federal schools. To obtain valid results and due to the impossibility of interviewing the entire population, the authors of this paper developed a random sample of the universe. In this way, during October 2020, an email was sent to all schools requesting teachers to answer a questionnaire (A appendix). The email addresses were obtained through the Internet. The result of this study was based on the answer of 116 teachers, being 81 (69.2%) females and 35 (30.2%) males and it was considered significant according to the universe. The research question plans to verify if the students’ scholar and learning performance were affected during the COVID-19 pandemic in public and private schools of the city of Rio de Janeiro (Brazil) with respect to elementary and middle schools.

The participants of this research had the following profile: age from 48 to 63 years (56.0%), time of experience well distributed in the range of 13 to 37 years (94.9%), is a classroom teacher (74.1%), teaches in municipal schools (42.0%) or state schools (35.7%), in high school (38.8%) or in elementary school I (29.3%), and a Portuguese, English, or Spanish teacher (43.1%). From the collected data, the authors of this paper built a database in an electronic spreadsheet, and it was analyzed by the SPSS program (Statistical Package for the Social Science), version 22.0, and by the Microsoft Excel 2007 application.

For the characterization of the respondent teachers, a variable results descriptive analysis was carried out through frequency distributions with the proportions of interest and calculation of appropriate statistics for quantitative variables (minimum, maximum, average, median, standard deviation, percentiles, and coefficient of variation - CV). The variability in the distribution of a quantitative variable was considered low if ?? <0.20; moderate if 0.20≤ ?? <0.40 and high if ?? ≥0.40. To check the association between two qualitative variables, comparing the frequency distribution of a qualitative variable in independent groups, the chi-square test was used. When the chi-square test was inconclusive, Fisher's exact test was used instead.

In the Inferential Analysis of Quantitative Variables, the hypothesis of normality of the distribution was verified by the Kolmogorov-Smirnov and Shapiro-Wilk tests. The distribution of a variable was considered normal when the two normality tests concluded this way. The student’s t-test was used for the distributions of two independent groups when the variable followed distribution was normal in all groups. For variables that had the hypothesis of normality rejected in at least one of the groups and for ordinal variables, the comparison of two independent groups was performed using the Mann-Whitney nonparametric test. The distributions of a quantitative variable from more than two independent groups were compared by ANOVA, when the variable under test followed a normal distribution in all groups, or by the Kruskal-Wallis test when the variable under test did not follow a normal distribution in all groups.

Two repeated measurements from the same respondent were compared in pairs using the Wilcoxon signed posts test, since in all cases at least one of the measurements did not follow a normal distribution. More than two repeated measurements by the same respondent were compared in pairs using the Friedman test, since in all cases at least one of the measurements did not follow a normal distribution.

To analyze whether the correlation between quantitative variables, Spearman's Order Correlation Coefficient was calculated. The significance of the Correlation coefficients was assessed by the Correlation Coefficient Test whereby a coefficient is significantly non-zero if the p-value of the Correlation Test is less than the level of significance. As for the strength or intensity of the correlation, in this work, the correlation between two variables was considered strong enough only if the correlation coefficient has an absolute value greater than 0.7 and a moderate correlation was considered if it had an absolute value greater than 0.5 and less or equal to 0.7.

All discussions about the significance tests were carried out considering a maximum significance level of 5% (0.05). Details of the proposed methodology of descriptive and inferential statistics can be found in Triola [65], Favero et al. [66], and Medronho et al. [67].

Reviewer 3 Report

Dear Authors,

I appreciate the great efforts that you have made in response to my questions and comments. The revision clarifies almost all the points I raised and helps me (and hopefully readers) understand the current manuscript.

Therefore, my recommendation is to publish in its present form.

Kind regards,

The Reviewer

Author Response

Dear Reviewer,

Thank you very much for your comment.

We are very grateful for your guidance and for publishing this paper.

Kind regards,

Ana